# Dynamic Organellar Mapping in yeast reveals extensive protein localization changes during ER stress

Anna Platzek[1], Klára Odehnalová[1], Julia P. Schessner [2], Georg H. H. Borner [1,2] ✉ & Sebastian Schuck [1] ✉

Sophisticated techniques are available for systematic studies of yeast cell biology. However, it remains challenging to investigate protein subcellular localization changes on a proteome-wide scale. Here, we apply Dynamic Organellar Mapping by label-free mass spectrometry to detect localization changes of native, untagged proteins during endoplasmic reticulum (ER) stress. We find that hundreds of proteins shift between cellular compartments. For example, we show that numerous secretory pathway proteins accumulate in the ER, thus defining the extent and selectivity of ER retention of misfolded proteins. Furthermore, we identify candidate cargo proteins of the ER reflux pathway, determine constituents of reticulon clusters that segregate from the remainder of the ER and provide evidence for altered nuclear pore complex composition and nuclear import. These findings uncover protein relocalization as a major aspect of cellular reorganization during ER stress and establish Dynamic Organellar Maps as a powerful discovery tool in yeast.

The budding yeast, *Saccharomyces cerevisiae*, is an important model organism for basic research. Many fundamental cell biological processes, such as transcription, translation and protein targeting, have been studied in yeast through systematic approaches[1–3]. So far, comprehensive analyses of protein localization have relied on high-throughput microscopy of large strain collections in which proteins are labeled with fluorescent tags[4–7]. These efforts have been complemented by automated image analysis tools[8,9]. As a result, it is now possible to compare protein localization under different conditions and thereby detect protein localization changes[10–12]. Nonetheless, such systematic investigations remain complex endeavors that require substantial resources and technical expertise. Moreover, microscopy-based approaches in yeast usually involve tags, which can affect the abundance of labeled proteins, alter their localizations and disrupt their functions.

An alternative approach for investigating protein localization on a proteome-wide scale is spatial proteomics by mass spectrometry-based protein profiling[13–15]. Here, cells are lysed mechanically and fractionated by centrifugation to partially separate cell organelles. Importantly, the aim of this subcellular fractionation is not to purify individual organelles. Rather, the purpose is to generate, for each organelle, a characteristic abundance profile across fractions. Proteins in each fraction are then identified and quantified by mass spectrometry, and proteins associated with the same organelle will have similar profiles. Based on these profiles and pre-defined compartment marker proteins, machine learning techniques assign proteins to subcellular compartments to obtain an 'organellar map' of the cell. Comparisons of maps generated under different conditions reveal protein localization changes. To date, a single study has applied organellar mapping to yeast[16]. This analysis investigated unperturbed cells and yielded high-confidence organelle assignments for 900 of the approximately 5400 proteins present in yeast under laboratory conditions[17]. No study in yeast so far has used organellar mapping to investigate protein localization changes.

Several protein profiling methods are available, which differ in the techniques for cell lysis and fractionation, mass spectrometry, and

[1]Heidelberg University Biochemistry Center, Heidelberg, Germany. [2]Department of Proteomics and Signal Transduction, Systems Biology of Membrane Trafficking Research Group, Max-Planck Institute of Biochemistry, Martinsried, Germany. ✉e-mail: borner@biochem.mpg.de; sebastian.schuck@bzh.uni-heidelberg.de

data analysis[18–23]. One such method is the 'Dynamic Organellar Maps' (DOMs) approach, which has been used extensively for comparative applications[18,24–26]. DOMs are based on cell fractionation by differential centrifugation and quantitative label-free mass spectrometry. The robustness and relative simplicity of these techniques ensures reproducibility and scalability, thus allowing comparisons of multiple samples. In addition, DOMs are supported by freely available online software for automated data analysis and visualization[26]. We therefore chose to adapt the DOMs approach for comparative organellar mapping in yeast.

To evaluate the utility of yeast DOMs, we investigated the cellular response to endoplasmic reticulum (ER) stress. ER stress is defined by an accumulation of misfolded newly synthesized proteins in the ER and can arise during metabolic fluctuations, adverse environmental conditions, cell differentiation and disease. ER stress activates the unfolded protein response (UPR), which enhances protein folding and the degradation of misfolded proteins[27]. In yeast, the UPR induces hundreds of genes, which encode ER-resident protein folding and degradation machinery but also proteins that localize to post-ER compartments of the secretory pathway, mitochondria, the cytosol and the nucleus[28]. Thus, ER stress alters the abundance of many proteins in various subcellular compartments. By contrast, high-throughput microscopy has identified a mere 25 proteins that showed localization changes during ER stress[11]. These results raise the question whether subcellular localization changes are a minor or an underappreciated aspect of cell adaptation to ER stress.

Here, we show that DOMs detect numerous alterations of protein transport processes that are known to be affected by ER stress and additionally uncover unexpected protein localization changes in processes that have not been linked to ER stress before. These findings provide a much expanded view of proteome remodeling during ER stress and demonstrate the power of DOMs for systematic investigations in yeast.

## Results

### Establishment of the DOMs spatial proteomics approach in yeast

We first generated steady-state maps of unperturbed yeast. Cells from a liquid culture were lysed by enzymatic cell wall removal and suspension in a hypo-osmotic buffer. Organelles were released by gentle mechanical sample homogenization, unbroken cells were removed by a clearing spin, and cell lysates were fractionated through a series of centrifugation steps to obtain material that sedimented at 1, 3, 6, 12, 24 or $78 \times 1000\,g$, or remained in the supernatant (Fig. 1a). We prepared six replicate maps, obtained in batches of two on three separate days. The total cell lysate and subcellular fractions were analyzed by quantitative label-free mass spectrometry. Protein intensity data from the total cell lysates were converted into abundance estimates in molecules per cell for 3910 proteins (Supplementary Data 1a). Another nearly 1500 proteins were previously detected in yeast but could not be quantified in our analysis. Many of these proteins are expressed at low levels[17], which likely explains why we could not obtain abundance estimates for them. The 78,000 g supernatant is called 'cytosol fraction', and we derived cytosolic pool estimates for each protein by dividing a protein's intensity in the cytosol fraction by the summed intensities in all organelle and cytosol fractions (Supplementary Data 1b). Of note, the cytosol fraction contained a minor proportion of soluble proteins that had leaked from the lumen of organelles and membrane proteins that presumably were part of small vesicles. In addition, many nuclear proteins showed substantial cytosolic pools, most likely because nuclei are mechanically damaged during sample preparation (Supplementary Fig. 1; Supplementary Data 1b). Conversely, cytosolic proteins may be found in the organelle fractions if they are part of large protein complexes or have membrane-associated pools. We processed the mass spectrometry data with the DOM-ABC

software[26] and identified 2971 proteins with reproducible abundance profiles across the six organelle fractions (Supplementary Data 1c; Supplementary Fig. 2a).

To enable an assignment of the profiled proteins to cell organelles, we next defined a set of compartment marker proteins. With the help of the Saccharomyces Genome Database and relevant literature, we built a reference database of the approximately 5400 proteins present in yeast under laboratory conditions[9,17]. Where available evidence allowed, we annotated these proteins with their predominant subcellular localization at steady state (Supplementary Data 2a). Using this information, we asked whether our experimental approach had introduced a bias by comparing the compartment distributions of the 2971 profiled proteins and all 5400 expressed proteins present in the reference database. The compartment distributions of the two sets of proteins were nearly identical (Supplementary Fig. 2b). Hence, our analysis overall faithfully represented the different subcellular compartments. Next, we used the reference database to iteratively train the support vector machine (SVM) module of DOM-ABC[26] to assign one of twelve possible subcellular localizations to the profiled proteins. The classifiers were nucleus, cytosol, mitochondria, vacuole (including endosomes), plasma membrane (including cell wall), Golgi, ER, ribosome, nuclear envelope, proteasome, lipid droplets and peroxisomes. Through this process, we retrieved 1908 proteins for which the SVM assignment matched the annotation in the reference database, and we used these proteins as compartment markers (Supplementary Data 2b). A simplifying visualization of the multi-dimensional map data in only two dimensions by principal component analysis (PCA) showed reproducible clustering of the compartment markers, indicating that the DOMs approach allowed robust organellar mapping in yeast (Fig. 1b; Supplementary Fig. 2c, d).

We then applied the trained SVMs to predict the localizations of all 2971 profiled proteins (Fig. 1c; Supplementary Data 1c). This prediction showed high recall and precision across compartments (Supplementary Fig. 2e). For 996 proteins, the SVM prediction did not match the annotation in the reference database. This incongruence mainly arose from two sources. First, 418 proteins had classifiers in the reference database that were not available to the SVMs, such as actin-associated, ambiguous or unknown. Second, the vast majority of the remaining 578 mismatched annotations concerned proteins that had large cytosolic pools in addition to organellar or nuclear pools and thus had multiple subcellular localizations[13]. In these cases, the SVM classification may not align with the majority localization in the reference database. When we disregarded proteins with classifiers unavailable to the SVMs and added 'cytosol' as a secondary annotation for proteins with cytosolic pool estimates >30%, the agreement between SVM predictions and the reference database increased to 89% (Supplementary Data 1d). Thus, despite limitations, the SVM predictions broadly captured the complexity of subcellular protein localization and provided a useful reference grid for the interpretation of protein localization changes.

In summary, we adapted the DOMs workflow to yeast and generated protein copy number estimates, cytosolic pool estimates and subcellular localization predictions. All data can be explored through an interactive tool, which also features a 'neighborhood predictor' to identify proteins with similar subcellular distributions (Supplementary Data 3).

### Application of DOMs to investigate the cellular response to ER stress

To test yeast DOMs in a comparative setting, we investigated protein localization changes elicited by ER stress. To induce misfolding of newly synthesized proteins in the ER, we applied two established ER stressors: dithiothreitol (DTT), which prevents disulfide bond formation, and tunicamycin, which blocks protein N-glycosylation. We fractionated untreated, DTT-treated and tunicamycin-treated cells as

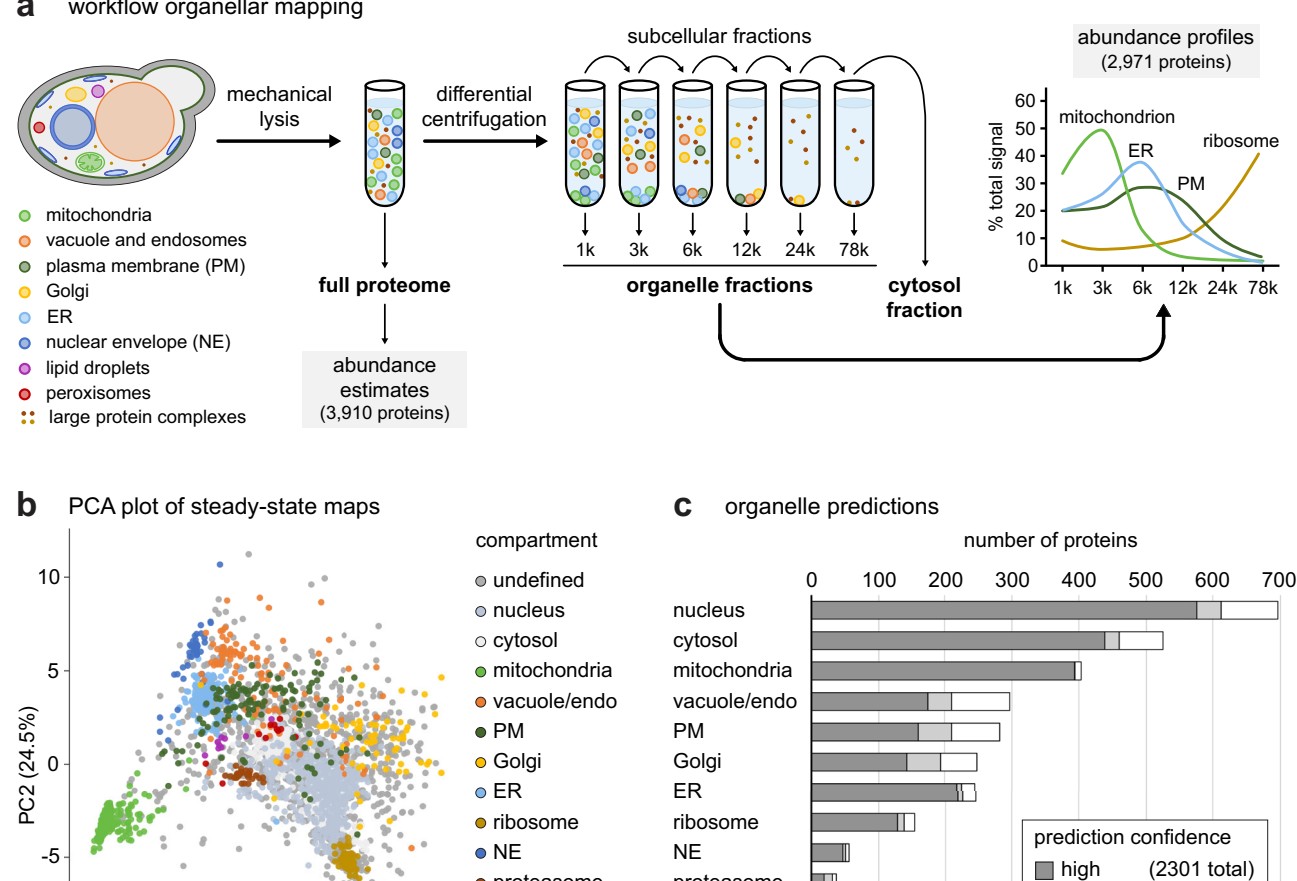

**Fig. 1 | Establishment of the DOMs spatial proteomics approach in yeast.**
**a** Workflow of organellar mapping. Cells are lysed mechanically, a full proteome sample is collected and the remainder of the cell lysate is subjected to differential centrifugation to obtain subcellular fractions, including a cytosol fraction. All samples are analyzed by quantitative label-free mass spectrometry, and an abundance profile is derived for each protein. Average profiles of proteins associated with particular organelles and ribosomes are shown to illustrate distinct fractionation behaviors and partial separation of subcellular compartments. **b** Principal component analysis (PCA) plot of steady-state organellar mapping data from unperturbed yeast. PCA was performed for the protein profiles in $n = 6$ maps obtained in biological triplicate with two technical replicates each. Pre-defined compartment markers are colored, other proteins are classified as 'undefined'. Supplementary Data 6 offers an interactive steady-state PCA plot, so that individual compartments can be selected or deselected. **c** Number of organelle predictions derived from steady-state maps with high, medium or low confidence. Source data are provided as a Source Data file.

above and analyzed fractions by quantitative mass spectrometry. As a prelude to organellar mapping, we determined the impact of DTT and tunicamycin on overall protein abundance. The two drugs caused distinct and reproducible proteome alterations, as indicated by PCA with global protein levels as input (Supplementary Fig. 3a). Volcano analysis revealed 1457 significantly changing proteins with DTT and 1183 with tunicamycin (Fig. 2a; Supplementary Data 4a, b). These numbers corresponded to about one-third of the measured proteomes, indicating major proteome remodeling. We had previously defined a set of core UPR target genes[29]. Of the corresponding proteins, most were upregulated by DTT and tunicamycin, demonstrating strong induction of ER stress (Fig. 2a). These proteins included ER-resident machinery for protein folding, modification and degradation, as well as machinery for protein transport into and out of the ER. In addition, numerous post-ER secretory pathway proteins were upregulated, as were components of the proteasome (Supplementary Fig. 3b–e). These observations agree with earlier data and reflect that cells respond to ER stress by adjusting protein secretion and degradation. Conversely, ribosomal proteins were slightly but consistently

downregulated, again in agreement with earlier data[28,29] (Supplementary Fig. 3f). Hence, DTT and tunicamycin caused many protein abundance changes expected to occur during ER stress.

DTT and tunicamycin cause ER stress by different mechanisms and may have additional effects unrelated to ER stress. Of the 1633 proteins differentially expressed in untreated and treated cells, 847 were shared between treatments, with highly correlated changes (Fig. 2b, c). 99% of the 847 shared hits showed changes in the same direction (i.e. upregulated or downregulated). Remarkably, the same was true for 88% of the 786 unique hits, i.e. proteins with abundance changes that were statistically significant in only one condition (Fig. 2d; Supplementary Data 4c, d). Hence, most differences between the effects of DTT and tunicamycin were quantitative rather than qualitative. Still, DTT caused a specific downregulation of mitochondrial electron transport chain proteins, while tunicamycin induced a small overall upregulation of mitochondrial proteins (Supplementary Fig. 4a, b). Thus, mitochondria appear to be differentially affected by the two treatments. Similarly, some integral plasma membrane transporters were upregulated by DTT but downregulated by tunicamycin

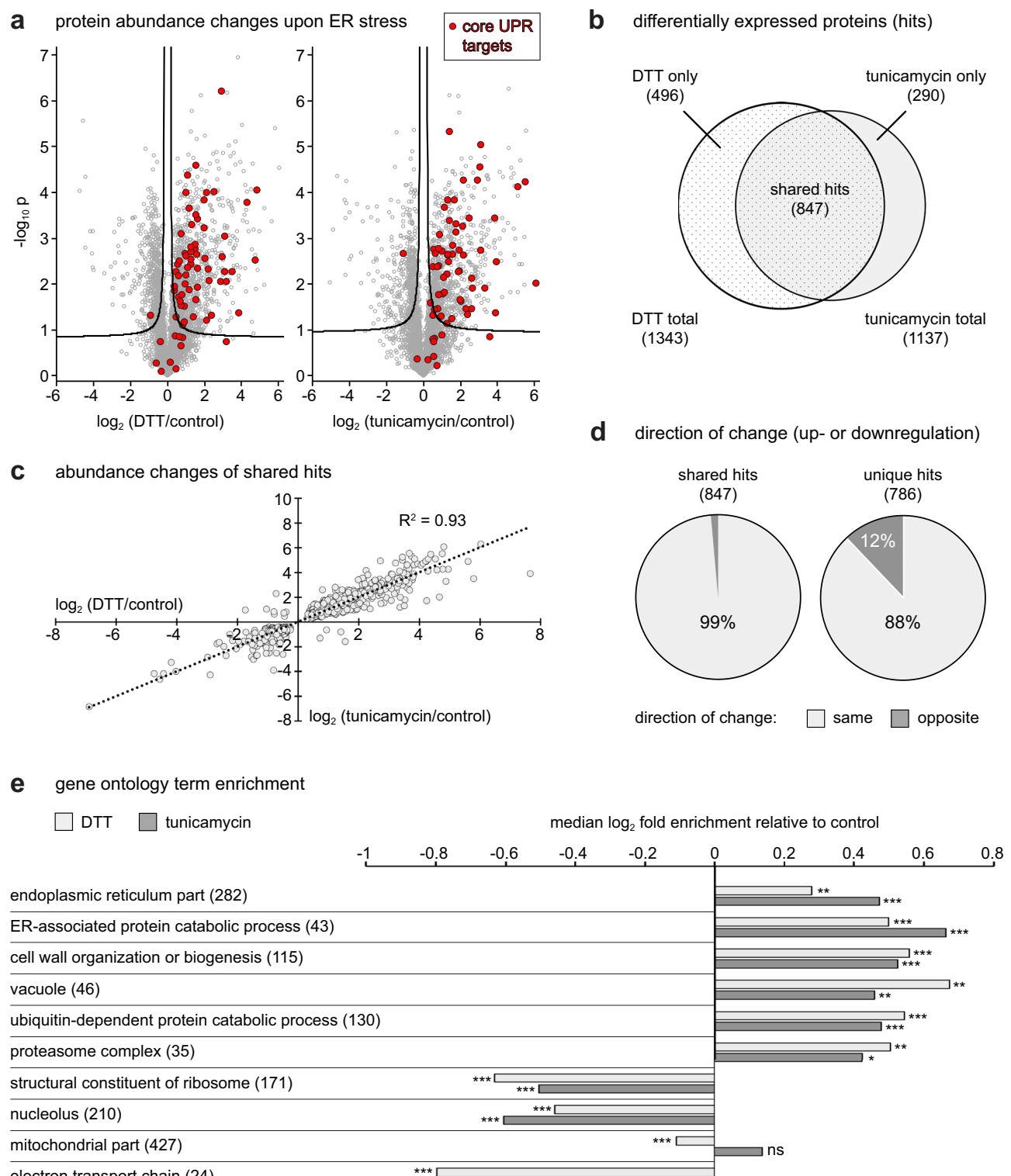

**Fig. 2 | Protein abundance changes upon ER stress. a** Volcano plots of full proteomes showing $\log_2$ fold abundance changes upon DTT or tunicamycin treatment. *P*-values for the significance of changes were calculated with a two-tailed t-test ($n=3$). Volcano lines indicate 5% false discovery rate cut-offs based on data permutation. Core UPR targets encoded by established UPR-regulated genes are highlighted in red. **b** Venn diagram of differentially expressed proteins upon DTT treatment, tunicamycin treatment, or both (shared hits). **c** Scatter plot of shared hits. Plotted are the $\log_2$ fold abundance changes upon DTT or tunicamycin treatment. **d** Pie charts showing the direction of change of shared hits and proteins that qualified as hits in only one stress condition (unique hits). **e** Gene ontology term enrichment analysis of DTT- and tunicamycin-treated samples versus control. Selected terms show shared up- or downregulation by both treatments as well as cases of divergent regulation. The number of proteins associated with each term is given in brackets. ns, not significant; *, false discovery rate (FDR) < 5%; **, FDR < 1%; ***, FDR < 0.1%. Source data are provided as a Source Data file.

(Supplementary Fig. 4c). A gene ontology term analysis confirmed these trends (Fig. 2e; Supplementary Data 4e, f). In sum, the global proteome data highlighted broad similarities and specific differences in the protein abundance changes caused by DTT and tunicamycin.

Next, we analyzed protein localization changes during stress. Importantly, the abundance profiles that form the basis for assessing protein localization are normalized (see methods). As a result, localization changes can be analyzed independently of abundance changes. Organellar maps of control, DTT-treated and tunicamycin-treated cells had comparable depths of around 3000 proteins (Supplementary Fig. 5a, b). PCA showed that gross map topology was similar, with three prominent exceptions (Fig. 3a). First, DTT caused a striking shift of the ER cluster, which was only weakly recapitulated with tunicamycin (light blue cluster in Fig. 3a, see Supplementary Fig. 5d for a simplified PCA plot). Second, the nuclear envelope cluster largely followed the ER shift with DTT, consistent with the physical continuity of ER and nuclear envelope (dark blue cluster in Fig. 3a; Supplementary Fig. 5d). Although the protein composition of the ER is similarly altered by DTT and tunicamycin (Fig. 2), DTT triggers a more extensive remodeling of ER morphology[30,31]. This stronger impact of DTT on ER structure may result in altered vesiculation of the ER membrane upon cell lysis, a different behavior during subcellular fractionation and, hence, the shift difference revealed by PCA. Third, tunicamycin but not DTT induced a shift of the mitochondrial cluster. This shift may be linked to the upregulation of mitochondrial protein abundance by tunicamycin (Supplementary Fig. 4b), but its physical basis is unclear. Statistical analysis based on the average abundance profiles of organelle markers extended the observations described above and showed that DTT induced significant overall shifts of ER, nuclear envelope, nuclear and Golgi proteins, whereas tunicamycin caused significant shifts of nuclear envelope, Golgi and mitochondrial proteins (Supplementary Data 5a).

We then compared control and treatment maps with the established 'MR' analysis to identify proteins with altered localizations[26]. This analysis is based on a comparison of a protein's abundance profiles under control and treatment conditions (Fig. 3b). Furthermore, it takes into account both the magnitude of a movement (M-score) and the reproducibility of the shift direction (R-score). At a false discovery rate of <5%, we identified 410 candidate relocalizing proteins with DTT and 149 with tunicamycin (Fig. 3c; Supplementary Data 5b, c). Cross-referencing with the whole proteome data revealed that the majority of these hits did not change in abundance (Supplementary Fig. 5c; Supplementary Data 5d). Hence, a sizeable fraction of ER stress-responsive proteins are regulated spatially rather than by an abundance change. Analysis at the organelle level showed that DTT hits included most of the ER proteins present in the maps (Fig. 3d; Supplementary Data 5b; 171 out of 183 mapped ER proteins). This observation was consistent with the finding that the entire organelle had an altered behavior in the subcellular fractionation after DTT treatment. After excluding ER proteins, DTT and tunicamycin hits had broadly similar distributions across the remaining organelles (Fig. 3d).

These data further confirmed that DTT and tunicamycin had similar effects on proteome organization. Nevertheless, DTT affected many more proteins significantly, and with higher M-scores (Fig. 3c). In addition, the pronounced shifts of the ER and nuclear envelope clusters with DTT were valuable diagnostic markers for analyses of individual proteins that moved away from or towards these clusters (see below). We therefore focused our subsequent investigations on the effects of DTT.

To facilitate in-depth analyses of individual protein localization changes, we compiled all abundance and mapping data into the interactive ER Stress Maps Analysis Tool (Supplementary Data 6). For a given query protein, this tool provides abundance changes, cytosolic pool changes, localization predictions, shift analyses, configurable PCA and profile plots, and protein neighborhood analyses. The cytosolic pool estimates can be used to identify proteins that shift between the organelle fractions and the cytosol fraction (Supplementary Data 5e). The different database outputs allow predictions of potential stress-induced protein localization changes when interpreted jointly (see 'example proteins' tab in Supplementary Data 6 for illustration).

Most ER and nuclear envelope proteins showed a common profile shift (as illustrated in Fig. 3b), which allowed us to distinguish qualitatively two types of hits: those that followed the overall shift of the organelle containing the protein and those that reflected shifts distinct from the overall organelle behavior. We thus annotated the 410 DTT hits and categorized them into seven groups: ER proteins shifting with the whole organelle (169 proteins), proteins shifting away from the ER (12), proteins shifting towards the ER (89), nuclear envelope proteins shifting with the whole organelle (22), proteins shifting away from the nuclear envelope (17), proteins shifting away from mitochondria (7), and other localization changes (94) (Fig. 4a; Supplementary Data 7). For further analysis, we focused on the 219 proteins that showed altered distributions between compartments. For 113 of these, we were able to predict both the predominant steady-state localization and a different destination compartment under ER stress conditions, and hence the direction of a shift between two compartments (Fig. 4b, c).

### Experimental evaluation of candidate relocalizing proteins

We next tested the validity of the above analysis by microscopy. For this purpose, we fused candidate relocalizing proteins to fluorescent proteins through chromosomal gene tagging. Depending on the protein analyzed, we positioned the tag such that determinants important for correct localization remained intact, such as signal sequences for ER import, ER retrieval sequences, transmembrane domains of tail-anchored proteins, and glycosylphosphatidylinositol (GPI) anchor attachment sequences. For some candidates, the tag had to be placed into the lumen of an organelle, either because the tagged protein was entirely lumenal or because tagging of a transmembrane protein at its cytosolic terminus would have caused mislocalization. In these cases, we tagged proteins with superfolder GFP (sfGFP), which matures more effectively in the oxidizing interior of the ER than other fluorescent proteins, including the original GFP[32].

### Proteins moving away from the ER

The 12 proteins moving away from the ER significantly shifted towards the cytosol upon both DTT and tunicamycin treatment (Fig. 5a; Supplementary Data 7). Five of these were soluble lumenal ER proteins, namely Cpr5, Sil1, Fpr2, Pdi1, and Kar2. A possible route for this redistribution is a process called ER reflux, which involves the stress-induced relocalization of correctly targeted soluble proteins from the ER lumen to the cytosol. This reverse translocation is functionally enigmatic and has been analyzed mainly with artificial reporters. However, it has also been demonstrated for the endogenous proteins Cpr5 and Pdi1[33,34]. Of the over 300 ER proteins present in yeast, only 19 are lumenal (Supplementary Data 2c), and the proteomic data permitted an analysis of all of them. Besides the five proteins mentioned above, only Lhs1, Eug1, Scj1, and Mpd1 showed weak evidence for redistribution to the cytosol. The remaining ten lumenal proteins had no detectable cytosolic pools (Fig. 5b). This selectivity indicated that the ER membrane remains intact during cell fractionation. Fluorescence microscopy confirmed that sfGFP-labeled Sil1 and Pdi1 redistributed to the cytosol during DTT treatment, whereas Ero1 did not (Fig. 5c; Supplementary Fig. 6a–c). Overall Ero1 levels rose 8-fold during stress and thus more steeply than the levels of, for example, Cpr5, Fpr2, Pdi1l, and Kar2 (Supplementary Data 4a). Hence, ER import of newly synthesized Ero1 remained intact during stress, arguing against the possibility that the cytosol shifts of Cpr5, Sil1, Fpr2, Pdi1, and Kar2 were due to a general defect in protein targeting to the ER. Overall, DOMs indicate that a specific subset of ER lumenal proteins

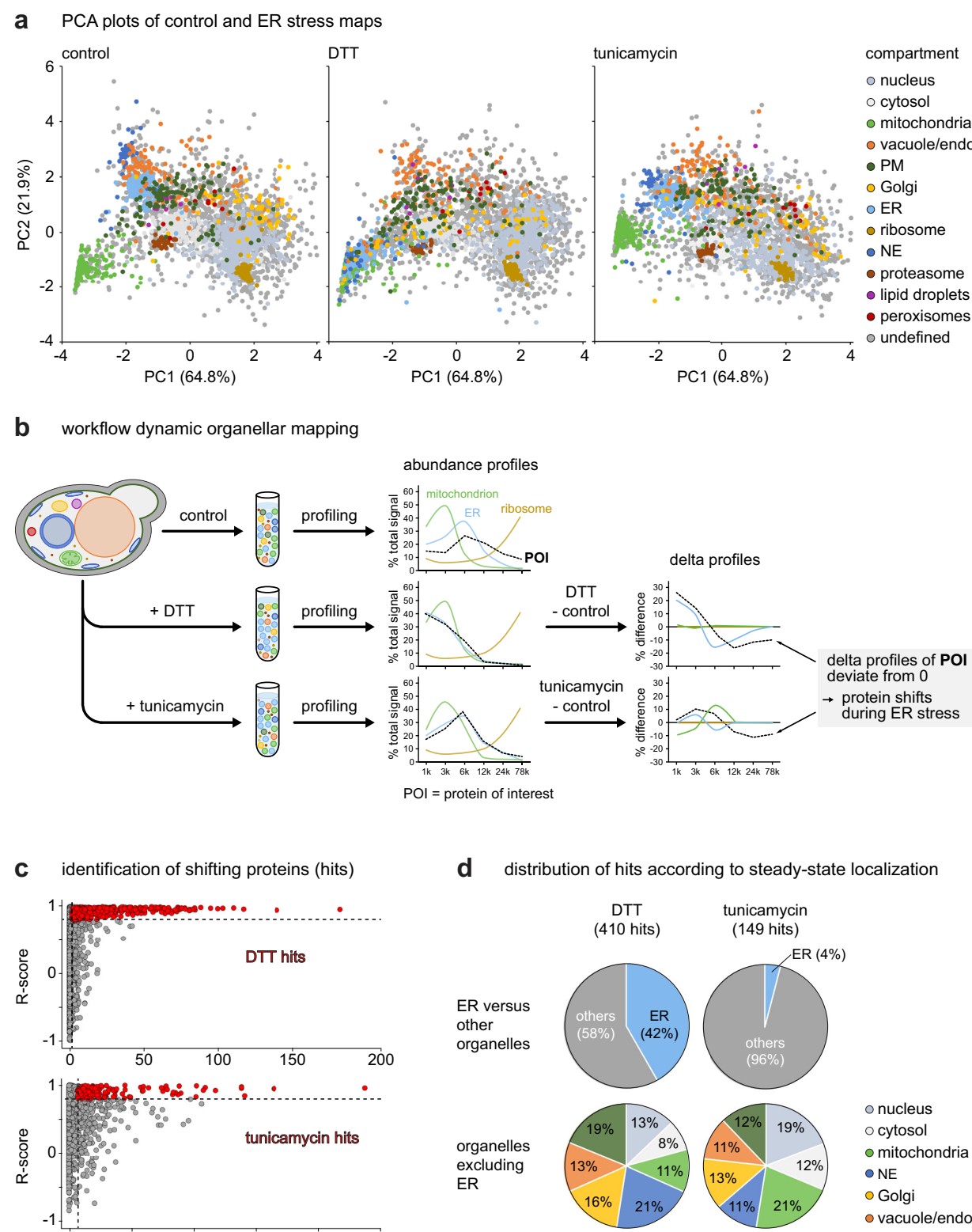

**a** PCA plots of control and ER stress maps

**b** workflow dynamic organellar mapping

POI = protein of interest

**c** identification of shifting proteins (hits)

**d** distribution of hits according to steady-state localization

relocalizes to the cytosol and thus provide new candidates for ER reflux cargos.

Besides the lumenal ER proteins discussed above, four peripheral ER membrane proteins (Get3, Swh1, Osh6, Far8) and three integral ER membrane proteins (Rtn1, Yop1, Dpm1) shifted towards the cytosol (Fig. 5a). A cytosolic shift of integral membrane proteins was unexpected. PCA showed that Rtn1, Yop1, and Dpm1 segregated from the ER

upon stress and clustered tightly, suggesting that they undergo similar localization changes (Fig. 5d). We found previously that Rtn1 forms punctate structures in about 30% of cells upon DTT treatment, and similar structures have been observed for a Yop1 homolog in fission yeast[35,36]. Rtn1 and Yop1 are reticulon homology domain proteins important for ER morphogenesis[37]. They interact with each other and with the tail-anchored protein Dpm1[38]. Four-color imaging showed that

**Fig. 3 | Protein localization changes upon ER stress. a** Principal component analysis (PCA) plot of control and ER stress maps. Pre-defined compartment markers are colored, all other proteins are classified as 'undefined'. Plots are based on the averaged profiles of $n$ = 3 biological replicates. The most pronounced change is the shift of the ER and NE clusters (light and dark blue), which obscure the mitochondria cluster (green) in the DTT map. Supplementary Data 6 offers an interactive version of these plots, so that individual compartments can be selected or deselected. **b** Workflow of dynamic organellar mapping, illustrating the derivation of delta profiles from the comparison of abundance profiles obtained under different conditions. **c** Identification of shifting proteins (hits) upon DTT or tunicamycin treatment by means of movement (M) and reproducibility (R) scores. Proteins highlighted in red are candidate hits with M-scores >1.3 and R-scores >0.8, with an estimated false discovery rate of <5%. **d** Pie charts showing organelle distributions of DTT and tunicamycin hits according to their steady-state localization, either as ER versus other organelles (top) or among organelles excluding the ER (bottom). Source data are provided as a Source Data file.

Rtn1, Yop1, and Dpm1 indeed co-localized in DTT-induced puncta, which segregated from the general ER marker Ubc6 (Fig. 5e). Tunicamycin also induced these puncta, albeit less efficiently (Supplementary Fig. 6e). Of note, the Rtn1-mCherry fusion protein remained intact under all conditions (Supplementary Fig. 6d). The nature of the structures underlying the stress-induced puncta remains to be clarified. In particular, it will be interesting to determine whether Rtn1, Yop1, and Dpm1 are part of a stress-induced ER subdomain or detach from the ER during stress. Either scenario could explain their shift to the cytosol fraction, which likely also contains small membrane vesicles that form in intact cells or arise through organelle fragmentation during cell lysis. In any case, DOMs identified new components of a largely uncharacterized ER subdomain or ER-derived structure.

## Proteins redistributing towards the ER

It has long been appreciated that newly synthesized proteins that fail to fold properly after import into the ER are retained there, thus ensuring that only functional proteins are supplied to post-ER compartments[39,40]. The stressors DTT and tunicamycin cause pervasive protein misfolding and are expected to lead to ER retention of many proteins. The DOMs approach made it possible to test this expectation on a proteome-wide scale. To identify proteins that redistributed to the ER during stress, we searched for candidates whose abundance profiles across subcellular fractions became more similar to the average profile of ER marker proteins. The large baseline shift of the ER cluster upon DTT treatment strongly enhanced the predictive power of this criterion. In addition, we required that candidates were also flagged as hits in the MR analysis and had a known or predicted post-ER secretory pathway localization. This analysis yielded 89 hits (Supplementary Data 7). In untreated cells, 7 of these localized to the ER-Golgi intermediate compartment (ERGIC), 33 to the Golgi, 29 to the plasma membrane or cell wall, 13 to the vacuole or endosomes, and 7 had an unclear post-ER localization in the secretory pathway. Of these 89 hits, 86 had predicted transmembrane domains or ER-targeting signal peptides and likely are integral membrane or lumenal proteins. The mapped and MR-analyzed proteins in DTT-treated cells included 243 integral membrane or lumenal proteins that localize to post-ER secretory pathway compartments (Supplementary Data 5g). These proteins must therefore pass through the ER. The 86 transmembrane or lumenal proteins shifting to the ER are a minority among these 243 proteins. This fact implies that there was no general ER export block during stress, but that certain proteins traversing the ER were particularly liable to misfolding and retention.

The 13 vacuole or endosome proteins shifting towards the ER included all five soluble proteins that were present in the maps and are targeted to the vacuole lumen via the CPY pathway (Prc1/Cpy, Pep4, Atg42, Ape3, Ydr415c)[41]. In addition, they included two membrane proteins that act as sorting receptors in the CPY pathway (Vth1 and Vth2)[42]. Pep1/Vps10, the third sorting receptor of the pathway, cycles between the Golgi and endosomes[43], and it also shifted towards the ER upon stress (Fig. 6a; Supplementary Data 7). By contrast, transport of vacuole membrane proteins along the ALP pathway appeared to remain intact because none of its prototypical cargos (Pho8, Nyv1, Yck3, Atg27)[41] changed their predicted localization (Supplementary Data 5b). Prc1 shifted particularly strongly as its abundance profile correlated well with the average profile of vacuole marker proteins under control conditions but aligned with the average profile of ER marker proteins under stress (Fig. 6b). Microscopy confirmed stress-induced redistribution of Prc1, Atg42, and Pep1 towards the ER upon both DTT and tunicamycin treatment (Fig. 6c; Supplementary Fig. 7a-d). An imaging pulse-chase experiment in which Prc1-sfGFP expression was induced in unstressed or ER-stressed cells showed that Prc1 synthesized during DTT treatment was strongly retained in the ER (Fig. 6d; Supplementary Fig. 7e). Finally, Western blotting revealed that Prc1-sfGFP was extensively cleaved at steady-state so that free sfGFP was generated, most likely by proteolytic processing in the vacuole (Fig. 6e). Cleavage was less extensive during ER stress, presumably because ER-retained Prc1-sfGFP was not cleaved. Hence, microscopy conflates the distribution of full-length Prc1-sfGFP and free sfGFP. This technical shortcoming is circumvented by dynamic organellar mapping, which analyzes native Prc1. Overall, these results indicate that ER stress impairs the targeting of vacuole proteins by causing misfolding and ER retention of cargos and sorting receptors, particularly of the CPY pathway.

The 29 plasma membrane or cell wall proteins with stress-induced shifts towards the ER included 11 predicted or validated GPI-anchored proteins (out of 13 mapped GPI-anchored proteins)[44,45]. GPI-anchored proteins are exposed to the extracellular space and are expected to be post-translationally modified. It is therefore plausible that their maturation requires the ER-resident protein folding and modification machinery, which becomes overwhelmed during stress. Gas1 and Gas3 were two GPI-anchored proteins that showed large shifts towards the ER cluster in PCA plots (Fig. 7A). Microscopy confirmed their stress-induced redistribution towards the ER (Fig. 7b; Supplementary Fig. 8a–c). In addition, we tested Utr2, which was one of the two GPI-anchored proteins that had not satisfied all of our criteria for proteins shifting towards the ER (Supplementary Data 5b). Nevertheless, Utr2 showed a moderate shift in PCA plots (Fig. 7a) and appeared in the ER upon DTT treatment (Fig. 7b, note the Utr2 signal at the nuclear envelope). Another imaging pulse-chase experiment showed that ss-sfGFP-Gas3 synthesized during ER stress was largely retained in the ER (Supplementary Fig. 8d). These results suggest that most GPI-anchored proteins were subject to ER retention during stress.

Next, we validated ER redistribution of Golgi proteins. Among the 33 proteins suggested by the data analysis were 19 mannosyltransferases (out of 20 profiled Golgi-localized mannosyltransferases[46]; Fig. 7c). We confirmed the predicted DTT-induced ER relocalization for Mnn2 and Mnn5 (Fig. 7d; Supplementary Fig. 8e, f). By contrast, and as predicted by the maps, the Golgi membrane protein Aur1 did not relocalize, showing that only specific Golgi proteins accumulated in the ER (Fig. 7d). It remains to be determined whether mannosyltransferases accumulate in the ER because of misfolding of newly synthesized molecules or because they naturally cycle between the ER and the Golgi, and stress shifts the transport equilibrium towards the ER[47].

Taken together, DOMs identified many secretory pathway proteins that accumulated in the ER, either due to misfolding and ER retention of newly synthesized molecules or due to trapping of pre-existing molecules that normally cycle within the early secretory pathway. These results agree with prior knowledge of ER protein quality control and reveal the extent to which stress alters protein transport. Furthermore,

**a**   Classification of predicted shifts of DTT hits (410)

**b**   Predicted shifts between compartments (113)

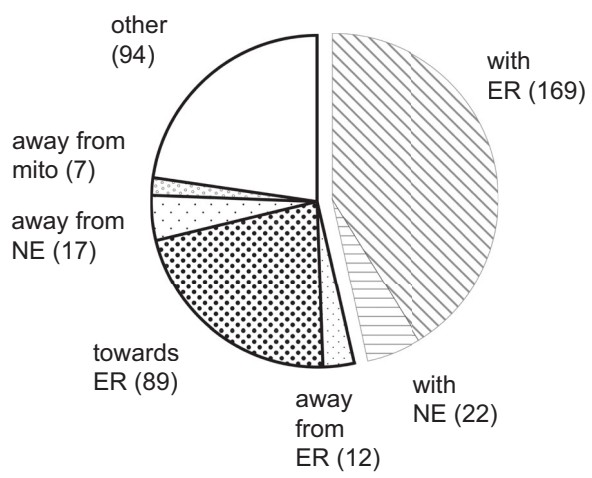

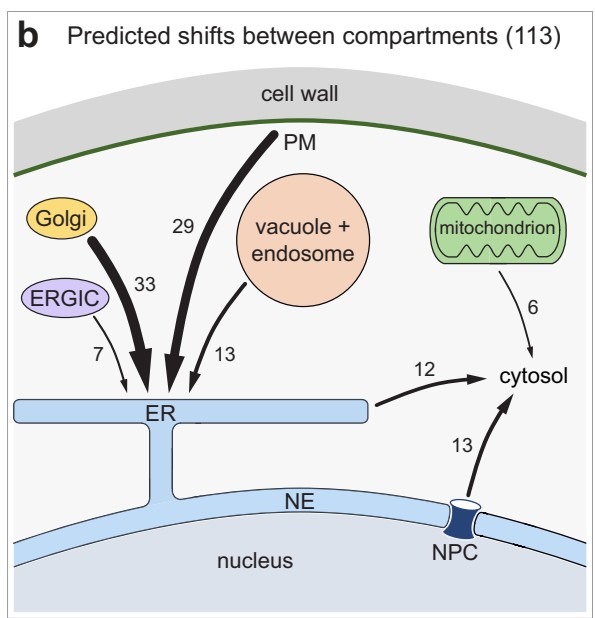

**c**   Summary of DTT hits with predicted shifts between compartments

| predicted shift | origin | destination | proteins | description |
|---|---|---|---|---|
| away from ER | ER lumen | cytosol | Cpr5, Fpr2, Kar2, Pdi1, Sil1 | ER reflux |
| | ER periphery | cytosol | Far8, Get3, Osh6, Swh1 | diverse proteins |
| | ER membrane | cytosol | Dpm1, Rtn1, Yop1 | Rtn1 puncta |
| towards ER | plasma membrane, cell wall | ER | Ccw14, Crh1, Cwp1, Dfg5, Emc33, Exg2 Gas1, Gas3, Gas5, Pst1, Toh1 | GPI-anchored proteins |
| | | | Bgl2, Chs2, Dnf2, Epo1, Exg1, Flc1, Flc2, Flc3, Fre1, Kre6, Kre9, Pho87, Pdr12, Sln1, Scw4, Scw10, Sim1, Tos1 | diverse proteins |
| | vacuole/endosome | ER | Ape3, Atg42, Pep4, Prc1, Vth1/2, Ydr415c | CPY pathway |
| | | | Atg15, Cps1, Ecm14, Ecm38, Ppn1, Prb1, Ydr262w | diverse proteins |
| | Golgi | ER | Anp1, Hoc1, Kre2, Ktr1, Ktr3, Ktr4, Ktr6, Ktr7, Mnn1, Mnn2, Mnn5, Mnn9, Mnn10, Mnn11, Mnt3, Och1, Sur1, Van1, Yur1 | mannosyltransferases |
| | | | Cdc1, Dcr2, Drs2, Gda1, Gnt1, Eps1, Erv2, Lcb5, Lem3, Mnl2, Pep1, Psg1, Ste13, Tre2 | diverse proteins |
| | ERGIC | ER | Emp24, Emp47, Erp1, Erp2, Erv25, Erv41, Erv46 | ER-to-Golgi transport |
| away from NE | NE | cytosol | Dyn2, Gle2, Nic96, Nsp1, Nup49, Nup57, Nup82, Nup116, Nup159, Nup192 | nucleoporins |
| | | | Mex67, Kap95, Srp1 | nucleocytoplasmic transport |
| away from mitochondria | mitochondria | cytosol | Ccp1, Ptc5, Ptc7, Tim9, Tim10, Tim13 | diverse proteins |

**Fig. 4 | Predicted protein localization changes upon DTT-induced ER stress.**
**a** Pie chart showing classification of predicted shifts of DTT hits, including shifts with the whole ER or nuclear envelope (NE). The number of proteins per category is given in brackets. mito, mitochondria. **b** Illustration of predicted shifts between compartments for 113 DTT hits. ERGIC ER-Golgi intermediate compartment, PM plasma membrane, NPC nuclear pore complex. **c** Summary of DTT hits with predicted shifts between compartments. Source data are provided as a Source Data file.

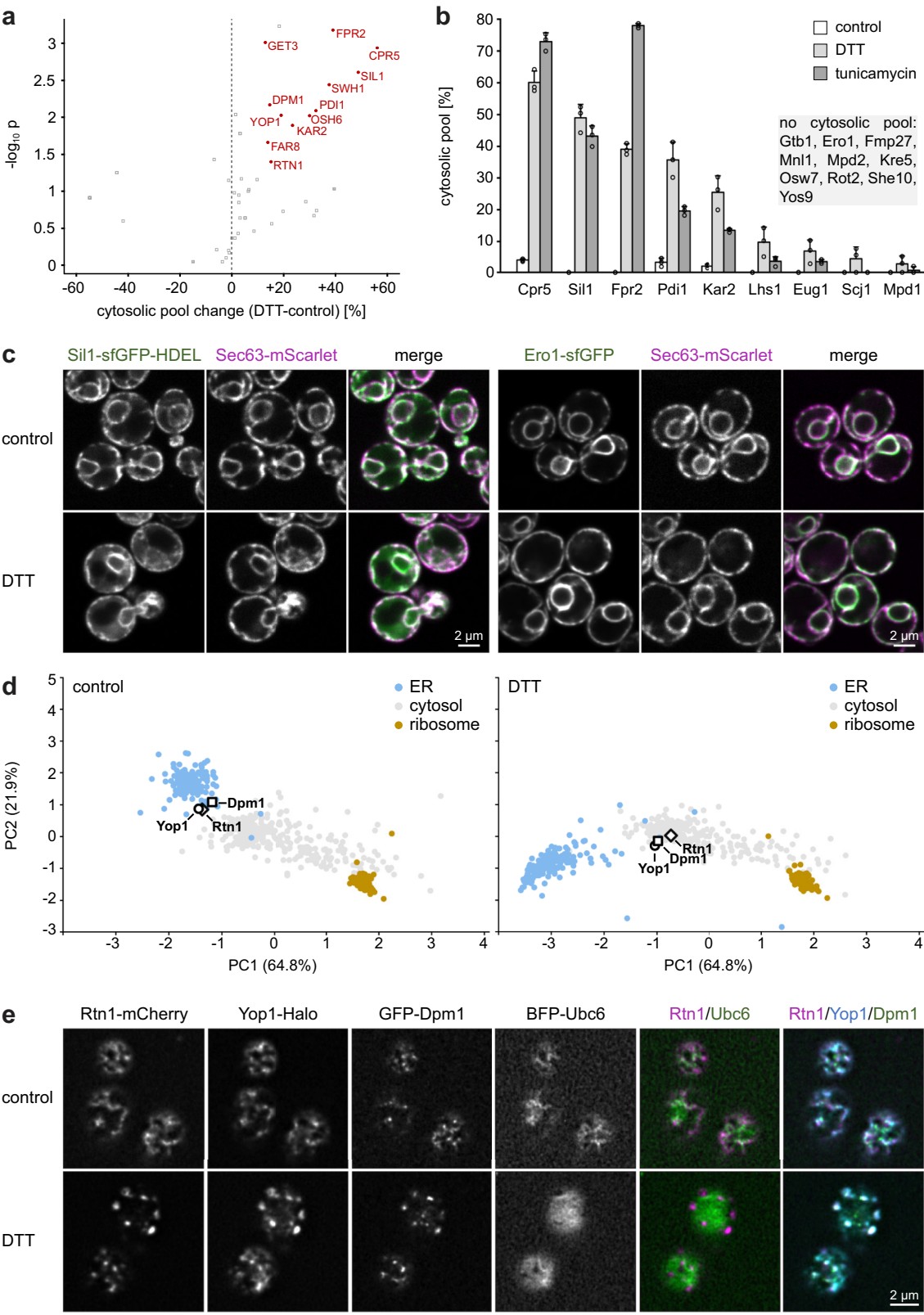

they indicate that specific classes of secretory proteins, namely lumenal vacuole proteins, GPI-anchored proteins, and Golgi-localized manno-syltransferases, are particularly liable to mislocalization.

## Redistribution of nucleoporins and importins

Finally, we turned to subcellular localization changes unrelated to the secretory pathway. The MR analysis indicated significant localization changes for many nuclear envelope proteins, including most components of the nuclear pore complex (called nucleoporins or Nups). However, a closer look at nucleoporins suggested a dichotomy. While 15 nucleoporins shifted with the main nuclear envelope cluster, 10 shifted away from it (Fig. 8a, compare Nsp1, Nup57, Nup82 with Nup84, Nup133, Nup170). Microscopy revealed that these divergent behaviors corresponded to distinct subcellular distributions, as Nsp1 formed

**Fig. 5 | Cytosolic redistribution of lumenal ER proteins and reticulon clustering upon ER stress. a** Volcano plot of the percentage point changes of the cytosolic pool of ER proteins upon DTT treatment. *P*-values for the significance of change were calculated with a two-tailed paired t-test ($n = 3$). Proteins with a consistent increase of their cytosolic pool of >10% with both DTT and tunicamycin treatment are highlighted in red. **b** Cytosolic pools of lumenal ER proteins in control, DTT-treated and tunicamycin-treated cells. Bars represent the mean of $n = 3$ biological replicates, error bars show standard deviations. **c** Confocal fluorescence images of mid sections of control and DTT-treated cells expressing the ER marker Sec63-mScarlet along with the lumenal ER proteins Sil1-sfGFP-HDEL or Ero1-sfGFP. Sil1 becomes partially cytosolic during ER stress, Ero1 remains ER-localized. **d** PCA plot of organellar maps of control and DTT-treated cells. Rtn1, Yop1, and Dpm1 shift away from the ER cluster upon DTT treatment. The cytosol and ribosome clusters do not shift and are shown for reference. Note, however, that the Rtn1, Yop1, and Dpm1 were identified as moving to the cytosol based on their increased cytosolic pools, as shown in (**a**). **e** Deconvolved confocal fluorescence images of cortical sections of control and DTT-treated cells expressing the ER marker BFP-Ubc6 along with Rtn1-mCherry, Yop1-Halo, and GFP-Dpm1. Rtn1, Yop1, and Dpm1 co-cluster upon DTT treatment. Source data are provided as a Source Data file.

cytosolic puncta upon ER stress whereas Nup84 did not (Fig. 8b; Supplementary Fig. 9a–c). We then imaged all 31 nucleoporins. Strikingly, the 10 nucleoporins predicted to shift away from the nuclear envelope formed cytosolic puncta, whereas the 15 nucleoporins predicted to shift with the nuclear envelope did not. Of the remaining six nucleoporins that were not DTT hits in the MR analysis, another two formed cytosolic puncta (Supplementary Data 8). Mapping these imaging results onto the PCA plot of DTT-treated cells revealed a clear separation of puncta-forming from non-puncta-forming nucleoporins (Fig. 8c). This separation corresponded well with the subcomplexes of the nuclear pore complex[48–50]. Specifically, components of cytoplasmic filaments and the channel complex as well as some linker Nups and inner ring components formed puncta. By contrast, components of the nuclear basket, membrane ring and outer ring did not form puncta (Fig. 8d, puncta-forming nucleoporins are underlined). Hence, DOMs accurately identified a subset of nucleoporins that partially redistributed to the cytosol upon stress.

Intriguingly, the puncta-forming nucleoporins Nup116 and Gle2 and the importin Kap95 showed very similar DTT-induced shifts in PCA plots, hinting at a shared localization change (Supplementary Fig. 9d). Nup116 and Gle2 interact, as do Nup116 and Kap95[51,52]. The common shift of Nup116 and Kap95 corresponded to the formation of cytosolic puncta containing both proteins (Fig. 8e; Supplementary Fig 9b, e). Given this cytosolic sequestration of the nuclear import receptor Kap95, we asked whether the transport of proteins with a nuclear localization sequence (NLS) was affected by ER stress. Indeed, nuclear import of newly synthesized NLS-mNeon was slowed (Fig. 8F; Supplementary Fig. 9f). In addition, some NLS-mNeon co-localized with Kap95 in cytosolic puncta (Supplementary Fig. 9g). By contrast, the nucleocytoplasmic distribution of blue fluorescent protein (BFP) without NLS was almost the same as in untreated cells, indicating that the observed import defect for NLS-mNeon was specific to active transport (Fig. 8f). These results illustrate that DOMs uncovered unexpected localization changes not previously associated with ER stress.

## Discussion

We have established Dynamic Organellar Maps (DOMs) in yeast and combined them with fluorescence microscopy to explore protein abundance and localization changes during ER stress. The use of DOMs as a guide for subsequent imaging experiments enabled a systematic investigation and showed that protein relocalization is a major aspect of cellular reorganization during ER stress. Our results indicate that (1) many secretory pathway proteins, in particular cargos of the CPY pathway for vacuolar protein sorting, GPI-anchored proteins and Golgi-resident mannosyltransferases, accumulate in the ER, at least in part due to misfolding and ER retention; (2) some but not all lumenal ER proteins redistribute to the cytosol, making them candidate ER reflux cargos; (3) a subset of integral ER membrane proteins, including reticulon homology domain proteins, co-cluster and segregate from the remainder of the ER; (4) specific nucleoporins and importins form common cytosolic clusters. Interestingly, we observed that many stress-responsive proteins show either abundance or localization changes, but less frequently both. This tendency has been noted before[12,23] and may reflect a more general regulatory principle.

Traditionally, microscopy and subcellular fractionation have been employed as parallel approaches to understand the spatial organization of cells. Since the advent of GFP, the imaging of fluorescently labeled proteins has become the predominant approach for analyzing protein localization in yeast[4,5,8,9]. However, recent advances in mass spectrometry, together with the realization that organelles need not necessarily be purified to determine their protein contents, have given rise to organellar mapping as another means of elucidating protein localization on a proteome-wide scale[13–15]. Indeed, our work indicates that the combination of DOMs and tailored follow-up microscopy studies offers several benefits.

First, DOMs detect native proteins, thus avoiding complications associated with tags such as GFP. Tail-anchored SNARE proteins, prenylated Rab GTPases, peroxisome proteins with classical targeting signals and ER lumenal proteins with retention signals all mislocalize when tagged at the C-terminus. Furthermore, many proteins with signal sequences for entry into the ER or mitochondria cannot reach their destinations when tagged at the N-terminus, and GPI-anchored proteins do not tolerate tags at either terminus. Accordingly, the five major candidate ER reflux cargos we identified show incorrect localizations in the yeast strain collection with C-terminally GFP-tagged proteins, and the eleven ER-retained GPI-anchored proteins show incorrect localizations in both the C- and N-terminal GFP fusion collections[5–7]. Therefore, these proteins could not be found in earlier work on stress-induced protein relocalization[11]. For many other proteins, it cannot be predicted a priori whether they retain their native localization after tagging[53].

Second, generating DOMs needs little equipment besides centrifuges and a mass spectrometer for label-free protein quantification. With our standard protocol, samples for up to four DOMs can be obtained in parallel by a single experimenter, and subsequent data analysis is aided by freely available software (https://domabc.bornerlab.org/QCtool)[26]. Reproducibility between biological replicates was high, and nearly 3000 proteins were mapped across three experimental conditions. We here used mass spectrometry with data-dependent acquisition. However, data-independent acquisition has been demonstrated to achieve a large increase in the number of mapped proteins in mammalian cells[26], and we anticipate that the same is true in yeast.

Third, DOMs yield predictions for the localization of uncharacterized proteins. There was 93% agreement of our localization assignments from steady-state maps with the only previous organellar mapping study in yeast[16] and 89% agreement with our literature-based reference database (Supplementary Data 1d, e). Furthermore, DOMs provided high-confidence predictions for 190 proteins with previously ambiguous or unknown localization (Supplementary Data 1f). It is important to bear in mind that many proteins have dual or multiple localizations[13], which the categorial single-organelle assignment for DOMs is not set up to accommodate. Therefore, new predictions need to be verified experimentally, and we emphasize that the main strength of DOMs is the detection of protein localization changes.

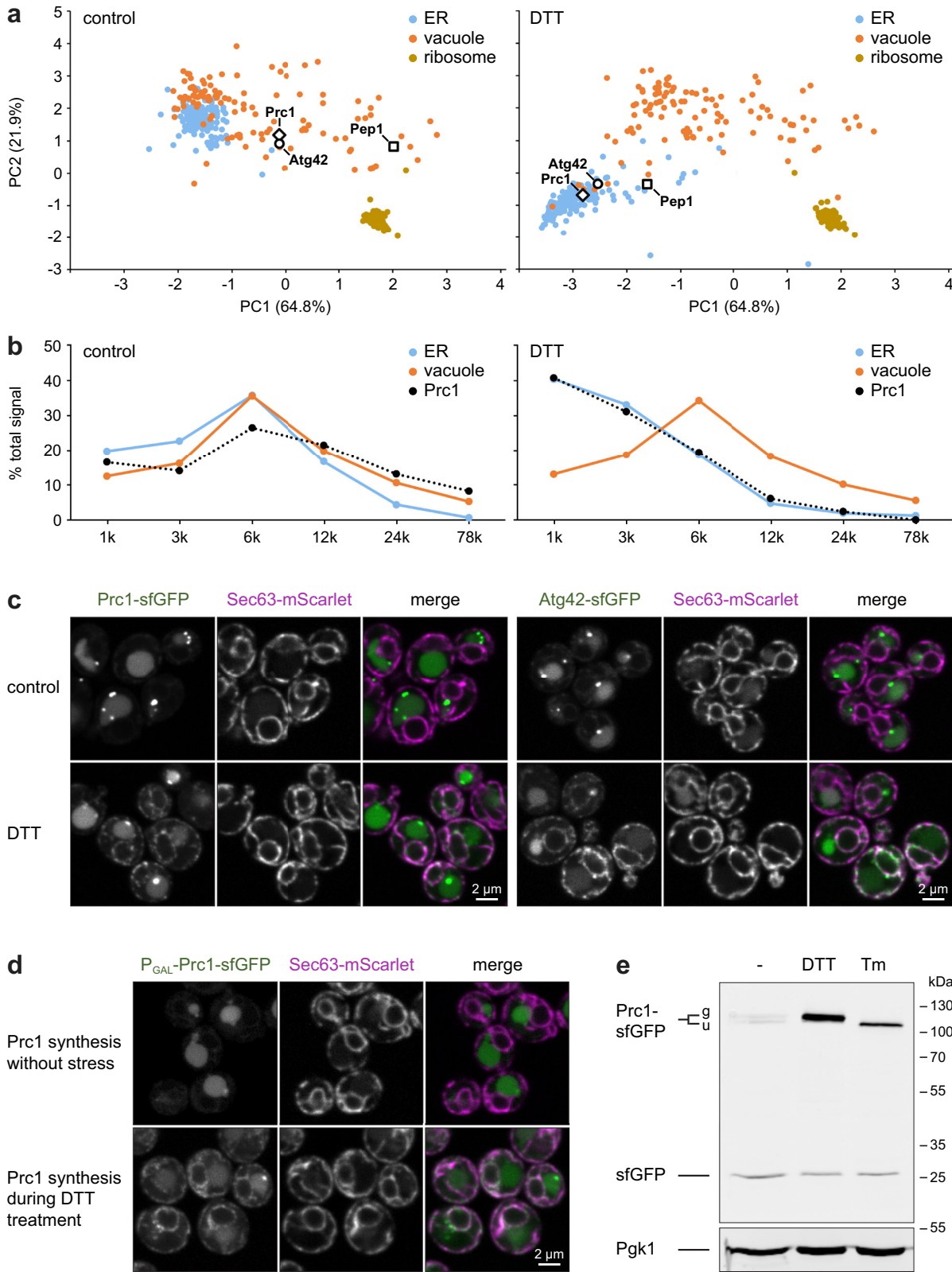

Various relocalizing proteins hint at previously unknown aspects of proteome remodeling during ER stress. For instance, cytosolic foci of nucleoporins have been found in yeast and metazoa under different stress conditions and may reflect the formation of protein condensates[54,55]. Our systematic analysis uncovered redistribution of numerous nucleoporins belonging to specific nuclear pore sub-complexes. One possibility is that cytosolic nucleoporin foci result

from pore complex assembly defects. Prolonged ER stress can overload the proteasome because of increased flux through the ER-associated degradation pathway and lead to misfolded protein accumulation and chaperone shortage in the cytosol[29]. Therefore, the integration of newly synthesized nucleoporins into nuclear pore complexes could fail, due to either their own misfolding or the lack of other factors essential for assembly[56]. Alternatively, cytosolic

**Fig. 6 | Redistribution of vacuole proteins towards the ER upon ER stress. a** PCA plot of organellar maps of control and DTT-treated cells. Prc1, Atg42 and Pep1 shift towards the ER cluster upon DTT treatment. The ribosome cluster is shown for reference. **b** Abundance profile of Prc1 and average abundance profiles of ER and vacuole marker proteins in control and DTT-treated cells. The Prc1 profile correlates with the vacuole profile in control cells and aligns with the ER profile upon DTT treatment. **c** Confocal fluorescence images of mid sections of control and DTT-treated cells expressing the ER marker Sec63-mScarlet along with the vacuole proteins Prc1-sfGFP or Atg42-sfGFP. Prc1 and Atg42 localize to endosomes (bright puncta) and the vacuole at steady state but redistribute towards the ER upon DTT treatment. **d** Confocal fluorescence images of cells expressing Prc1-sfGFP under an estradiol-inducible system based on the *GAL* promoter (P$_{GAL}$). Prc1 synthesis was induced in otherwise untreated cells (top) or during DTT treatment (bottom). **e** Western blot of GFP from untreated, DTT-treated, and tunicamycin (Tm)-treated cells expressing Prc1-sfGFP. g and u mark glycosylated and unglycosylated forms of Prc1-sfGFP. Tunicamycin prevents glycosylation of Prc1. Source data are provided as a Source Data file.

nucleoporin foci may arise from partial disassembly of pre-existing nuclear pore complexes. It has been proposed that the nucleoporin Nsp1, which forms pronounced foci upon DTT treatment, possesses chaperone activity and contributes to protein homeostasis in the cytosol[57]. Interestingly, we find that Get3, another protein with conditional chaperone activity[58], shifts from the ER membrane towards the cytosol during stress. These observations invite the intriguing speculation that stress-induced relocalization of Nsp1 and Get3 serves to increase cytosolic chaperone capacity. Moreover, in our ER-centric experimental validation, we did not follow up on several interesting predictions that concern other organelles (Supplementary Data 7). For example, a number of proteins shifted away from mitochondria (Ccp1, Msp1, Ptc5, Ptc7, Tim9, Tim10, Tim13). These shifts could be caused by disturbed mitochondrial protein import or stress-induced reflux of correctly targeted proteins. Some spindle pole body proteins moved with the nuclear envelope (Mps3, Cdc31) but others did not (Spc42, Spc110, Nud1, Cnm67), perhaps hinting at reorganization of the spindle pole body as a result of stress-induced cell cycle arrest. Finally, a number of organelle contact site proteins were predicted to relocalize upon ER stress (Efr3, Epo1, Ist2, Num1, Osh6, Scs2, Swh1, Tcb1/2/3). The biological significance of these predictions awaits elucidation.

In conclusion, DOMs are an accessible and powerful technique for unbiased, comprehensive studies of protein localization changes in yeast. We therefore anticipate that DOMs are going to join the growing arsenal of techniques for systematic analyses that make yeast such an illuminating model organism.

## Methods
### Plasmids
Plasmids and oligonucleotides used in this study are listed in Supplementary Data 9 and 10. To generate pFA6a-mScarlet-I3-HIS3 and pFA6a-mScarlet-I3-klTRP1, the mScarlet-I3 sequence[59] was amplified from a synthetic DNA fragment and inserted into pFA6a-GFP(S65T)-HIS3[60] or pFA6a-mNeonGreen-klTRP1, replacing GFP or mNeonGreen. To generate pFA6a-HaloTag-klTRP1, the HaloTag7 sequence[61] was amplified from a synthetic DNA fragment and inserted into pFA6a-mNeonGreen-klTRP1, replacing mNeonGreen. To generate pRS406-P$_{TEF}$-TagBFP-Ubc6, the *GPD* promoter of pRS406-P$_{GPD}$-mCherry-Ubc6 was replaced with the *TEF* promoter, creating pRS406-P$_{TEF}$-mCherry-Ubc6. Subsequently, TagBFP was amplified from pRS303H-P$_{GPD}$-TagBFP[62] and inserted into pRS406-P$_{TEF}$-mCherry-Ubc6, replacing mCherry. To generate pFA6a-hph-P$_{GAL}$-Kar2ss-sfGFP for N-terminal tagging of GPI-anchored proteins, the hygromycin resistance gene from pFA6a-hph was inserted into pFA6a-nat-P$_{GPD}$-yeGFP[63], creating pFA6a-hph-P$_{GPD}$-yeGFP. P$_{GPD}$-yeGFP was then replaced by P$_{GAL}$-Kar2ss-sfGFP amplified from pRS405-P$_{GAL}$-Kar2ss-sfGFP-HDEL. To generate pNH605-P$_{ADH}$-GEM-P$_{GAL}$-NLS-mNeonGreen, the mNeonGreen sequence was amplified from pFA6a-mNeonGreen-HIS3 while adding an N-terminal SV40 nuclear localization sequence and inserted into pNH605-P$_{ADH}$-GEM-P$_{GAL}$[29].

### Yeast strains
Strains used in this study were derived from *S. cerevisiae* W303 mating type a (SSY122) and are listed in Supplementary Data 11. Gene tagging was done with PCR products of the plasmids listed in Supplementary Data 9[60,63].

### Subcellular fractionation
Yeast (SSY122) were cultured at 30 °C in SCD medium containing 0.7% yeast nitrogen base (Merck), amino acids and 2% glucose. For steady-state maps, cultures were grown to early log phase (OD$_{600}$ = 0.1-0.5) and 200 ODs of cells were harvested by centrifugation at 3,000 g at room temperature for 5 min. For treatment maps, cultures were grown to mid log phase and either (1) diluted to OD$_{600}$ = 0.1, grown for 3 h and harvested, (2) diluted to OD$_{600}$ = 0.15, grown for 1 h, treated with 8 mM DTT (Roche), grown for another 2 h and harvested, or (3) diluted to OD$_{600}$ = 0.2, treated with 2 µg/ml tunicamycin (Merck), grown for another 3 h and harvested. Cells were resuspended to 20 ODs/ml in 10 ml spheroplast buffer (50 mM Tris-HCl pH 7.5, 1 M sorbitol, 0.5 mM MgCl$_2$). An aliquot of the cell suspension was diluted 1:40 in water and the OD$_{600}$ was measured, which gave a reading of about 0.5. To digest the cell wall, 40 µl zymolyase T100 (Biomol) were added to the cell suspension and cells were incubated at 30 °C for up to 20 min. During this time, spheroplast formation was monitored by periodically measuring the OD$_{600}$ as described above. Cell wall removal renders cells osmotically sensitive, leading to cell rupture and a drop in OD$_{600}$ when they are diluted in water. Once the OD$_{600}$ had fallen below 0.15, cell wall digestion was terminated by transferring the cell suspension onto ice. All subsequent steps were carried out in the cold. Spheroplasts were washed three times by pelleting at 1000 g for 2 min and gentle resuspension in 5 ml spheroplast buffer. After the last wash, cells were resuspended in 3.5 ml hypo-osmotic lysis buffer (25 mM Tris-HCl pH 7.5, 200 mM sorbitol, 1 mM EGTA, 0.5 mM MgCl$_2$, Roche protease inhibitors without EDTA) and sample volumes were adjusted to 4 ml with lysis buffer. Lysed spheroplasts were homogenized by 20 strokes in a Dounce homogenizer with a tight-fitting pestle (Thermo Fisher Scientific, catalog number 8853000007). Of the resulting homogenate, 50 µl were collected for full proteome analysis and combined with 12.5 µl 5x SDS resuspension buffer (50 mM Tris-HCl pH 8.1, 12.5% w/v SDS). DTT was added to a final concentration of 0.1 mM to prevent oxidation, samples were frozen in liquid nitrogen and stored at −80 °C.

The remainder of the homogenate was subjected to a clearing spin at 300 g at 4 °C for 5 min to remove unbroken cells. The supernatant was transferred into a 15 ml conical tube and centrifuged at 1000 × g at 4 °C for 10 min. The resulting 1k pellet was kept on ice, the supernatant was transferred into a new 15 ml conical tube and centrifuged at 3000 × g at 4 °C for 10 min. The resulting 3k pellet was kept on ice, the supernatant was transferred into an ultracentrifuge tube and centrifuged with a TLA-110 rotor in an Optima TLX tabletop ultracentrifuge (Beckman) at 6000 × g at 4 °C for 15 min. The resulting 6k pellet was kept on ice, the supernatant was transferred into a new ultracentrifuge tube and the procedure was repeated to obtain 12k, 24k and 78k pellets by successive centrifugation steps at 12,000 × g for 20 min, 24,000 × g for 20 min and 78,000 × g for 30 min. Of the 78k supernatant, 800 µl were mixed with 200 µl 5x SDS resuspension buffer (see above). All pellets were taken up in 1x SDS resuspension buffer. To ensure protein concentrations of at least 1 µg/µl, 200 µl 1x SDS resuspension buffer were used for the 1k pellet, 100 µl for the 3k, 6k,

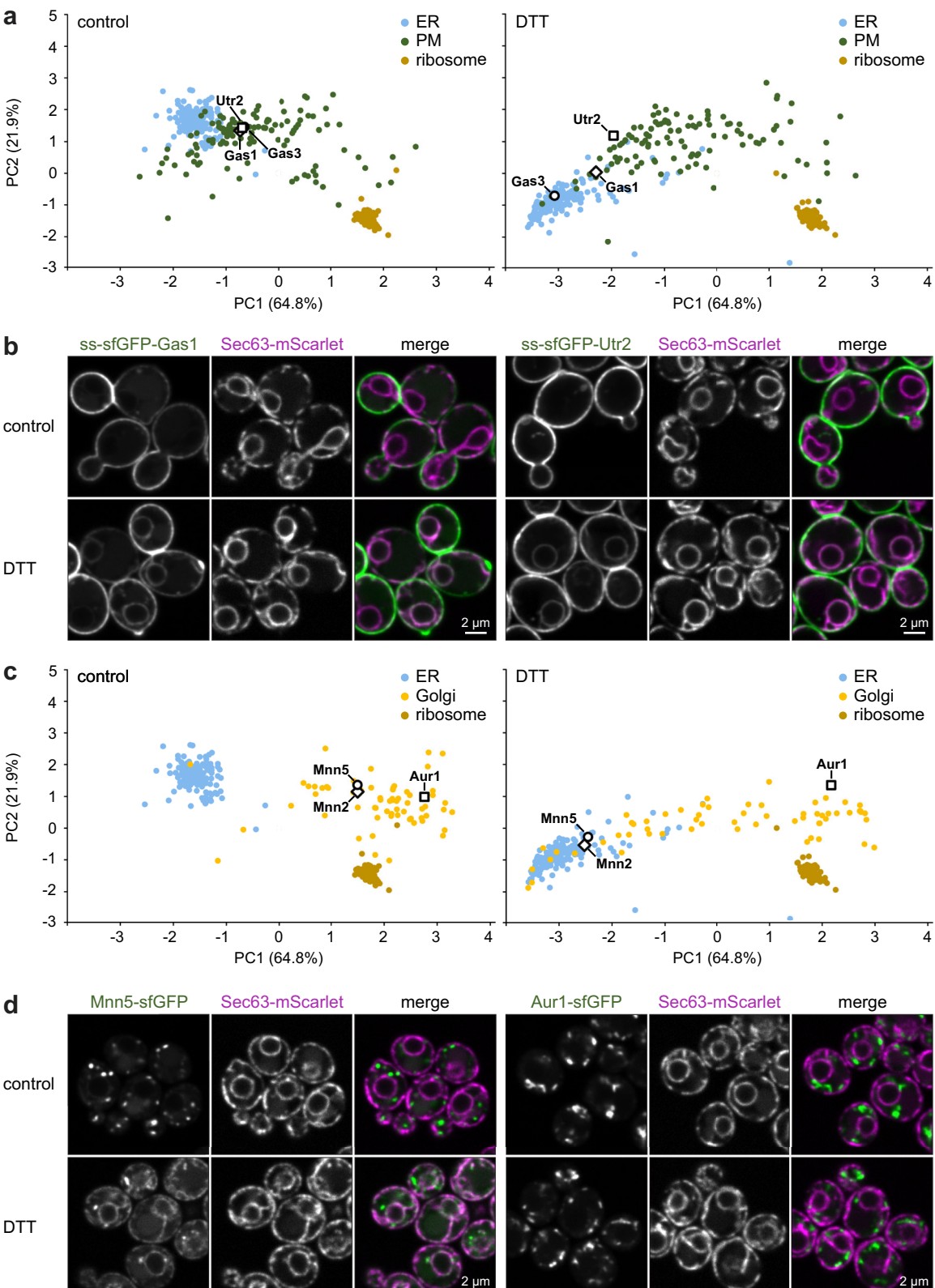

**Fig. 7 | Redistribution of plasma membrane and Golgi proteins towards the ER upon ER stress. a** PCA plot of organellar maps of control and DTT-treated cells. The GPI-anchored proteins Gas1, Gas3 and Utr2 shift towards the ER cluster upon DTT treatment. The ribosome cluster is shown for reference. **b** Confocal fluorescence images of mid sections of control and DTT-treated cells expressing the ER marker Sec63-mScarlet along with ss-sfGFP-Gas1 or ss-sfGFP-Utr2 (ss = signal sequence for ER targeting). Gas1 and Utr2 redistribute from the plasma membrane towards the ER upon DTT treatment. **c** PCA plot of organellar maps of control and DTT-treated cells. The Golgi proteins Mnn2 and Mnn5 but not Aur1 shift towards the ER cluster. **d** Confocal fluorescence images of mid sections of control and DTT-treated cells expressing the ER marker Sec63-mScarlet along with Mnn5-sfGFP or Aur1-sfGFP. Mnn5 but not Aur1 redistributes from the Golgi towards the ER upon DTT treatment. Source data are provided as a Source Data file.

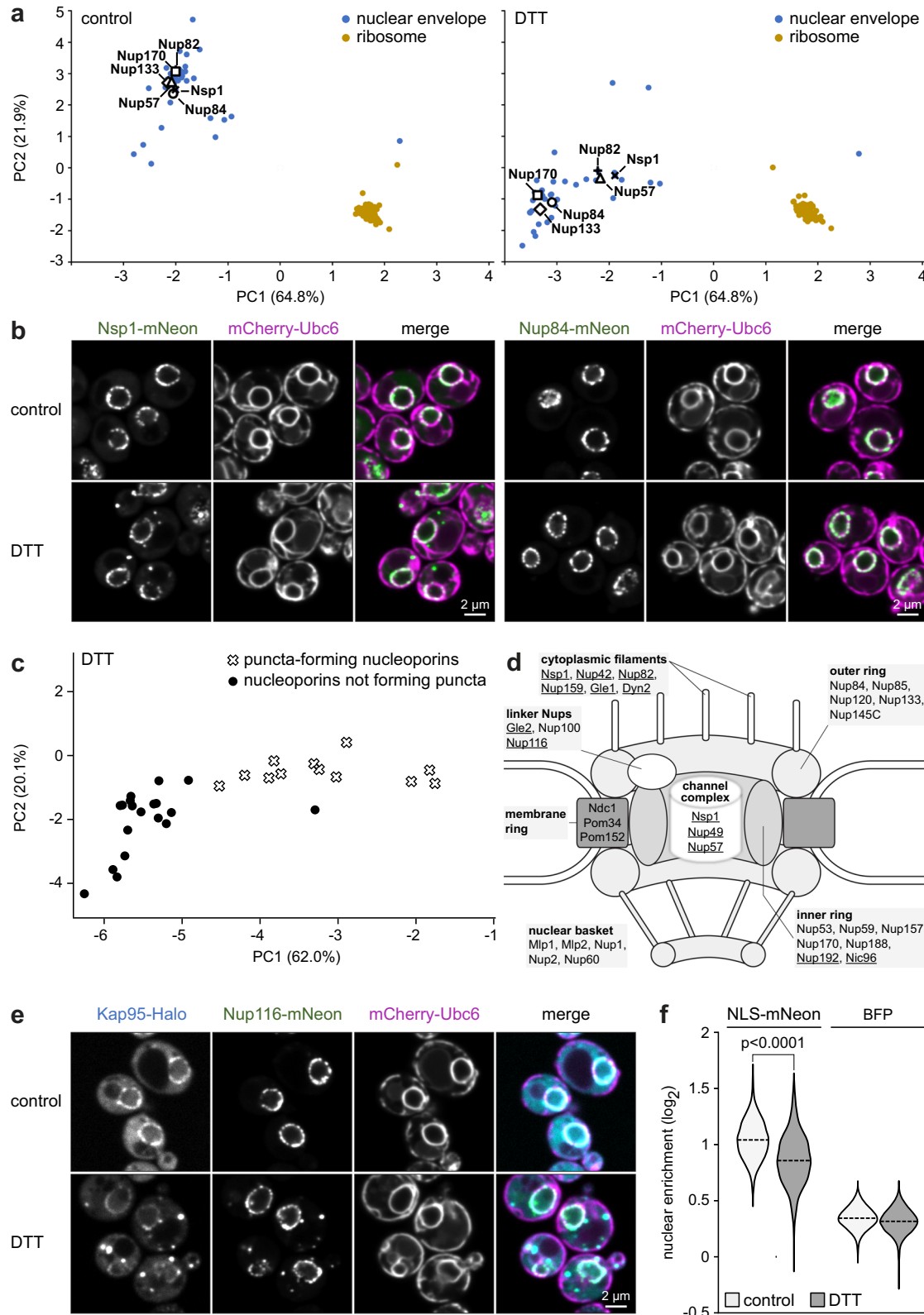

12k, and 24k pellets, and 150 µl for the 78k pellet. All samples were incubated at 65 °C for 3 min. The 1k pellet was further solubilized by 15 cycles of 30 s on/off at maximum intensity with a Bioruptor (Diagenode). DTT was added to all samples to a final concentration of 0.1 mM and samples were frozen in liquid nitrogen and stored at −80 °C.

**Peptide generation for mass spectrometry**

Proteins from total proteome, pellet, and 78k supernatant samples were converted into peptides for mass spectrometry as described[64]. Briefly, protein concentrations were determined with a BCA assay (Pierce) and samples were diluted to 1 µg/µl with 1x SDS resuspension buffer. Proteins were precipitated with ice-cold acetone, resuspended

**Fig. 8 | Impact of ER stress on the localization of nucleoporins and importins.**
**a** PCA plot of organellar maps of control and DTT-treated cells. Nucleoporins shift with the nuclear envelope cluster (e.g., Nup84) or segregate from it (e.g., Nsp1). The ribosome cluster is shown for reference. **b** Confocal fluorescence images of mid sections of control and DTT-treated cells expressing the ER marker mCherry-Ubc6 along with Nsp1-mNeon or Nup84-mNeon. Nsp1 forms cytosolic puncta upon DTT treatment, but Nup84 does not. **c** PCA plot of organellar maps of DTT-treated cells showing segregation of puncta-forming nucleoporins from nucleoporins that do not form cytosolic puncta. **d** Schematic architecture of the nuclear pore complex showing seven subcomplexes. Nucleoporins that form cytosolic puncta upon DTT treatment are underlined. **e** Confocal fluorescence images of mid sections of control and DTT-treated cells expressing the ER marker mCherry-Ubc6 along with the importin Kap95-Halo and the nucleoporin Nup116-mNeon. **f** Nuclear import of mNeon with a nuclear localization sequence (NLS-mNeon) expressed from an

inducible promoter system for 50 min. The plot shows the $\log_2$ fold enrichment of NLS-mNeon in nuclei versus whole cells in control and DTT-treated cells. Constitutively expressed blue fluorescent protein (BFP) without a targeting sequence served as a control for passive accumulation in the nucleus. Dashed lines indicate sample medians. The experiment was performed in biological triplicate and combined replicate data are shown with $n = 1142$ cells for NLS-mNeon control, $n = 859$ cells for NLS-mNeon DTT, $n = 1128$ cells for BFP control and $n = 1042$ cells for BFP DTT. The p-value was calculated with a two-tailed Mann-Whitney U-test. Nuclear enrichment of NLS-mNeon was reduced in DTT-treated cells ($p = 1.66 \times 10^{-82}$). Nucleo-cytoplasmic distribution of BFP was virtually unaffected by DTT, although the marginal difference between control and DTT-treated cells also reached statistical significance due to large sample sizes ($p = 1.35 \times 10^{-11}$). Source data are provided as a Source Data file.

in digestion buffer containing 8 M urea and 1 mM DTT, alkylated with 5 mM iodoacetamide and digested with LysC and trypsin. Peptides were acidified with trifluoroacetic acid, extracted with SDB-RPS Stop and Go Extraction (Stage) tips and eluted with 1.25% v/v ammonium hydroxide in 80% v/v acetonitrile. Sample volume was reduced to less than 5 µl with a vacuum Concentrator plus (Eppendorf) and peptides were resuspended in solution A (0.1% trifluoroacetic acid, 2% acetonitrile). Concentrations were determined with a NanoDrop 1000 spectrophotometer (Thermo Fisher Scientific) and adjusted to 150 ng/µl. Peptides were frozen in liquid nitrogen and stored at −80 °C.

## Mass spectrometry
Mass spectrometric measurements were done with data-dependent acquisition[26]. Nanoflow reversed-phase chromatography was performed with an EASY-nLC 1200 ultra-high-pressure system coupled to an Orbitrap Exploris 480 mass spectrometer via a nano-electrospray ion source (Thermo Fisher Scientific). A binary buffer system with mobile phases A (0.1% v/v formic acid) and B (0.1% v/v formic acid, 80% acetonitrile) was used. Peptides were separated in 110 min on a 50 cm × 75 µm (i.d.) column, packed in-house with ReproSil-Pur C18-AQ 1.9 µm silica beads (Dr. Maisch GmbH). The column was operated at 60 °C. Purified peptides were loaded onto the column in phase A and eluted with a linear 5-30% gradient of phase B, followed by washout and column re-equilibration. The mass spectrometer was controlled by Xcalibur software (Thermo Fisher Scientific) and operated in top 15 scan mode with a full scan range of 300−1650 Th. Survey scans were acquired at 60,000 resolution, with automatic gain control (AGC) set to 300% and a maximum ion injection time of 25 ms. Charge states were filtered for 2-5. Precursor ions were isolated in a window of 1.4 Th, fragmented by higher-energy collisional dissociation with normalized collision energies of 30%. Fragment scans were performed at 15,000 resolution with a maximum injection time of 28 ms, AGC set to 100% and a dynamic precursor exclusion for 30 s.

## Mass spectrometry raw data analysis
For protein identification, mass spectrometry raw data were analyzed in MaxQuant version 2.0.1.0[65]. Match between runs and label-free quantification (LFQ) were enabled. The minimum LFQ count was set to 1. Default parameters were used for all other settings. The MaxQuant experimental design restricted matching to equivalent and adjacent fractions within the six-fraction profiling speed gradient. Cytosol and full proteome fractions were only matched to equivalent fractions. For ER stress maps, matching was only possible within a treatment condition. Spectra were searched against the SwissProt FASTA *Saccharomyces cerevisiae* database downloaded from UniProt (6750 entries). The resulting intensity data and normalized profiles are provided as Supplementary Data 12 and 13. The mass spectrometry proteomics data have been deposited to the ProteomeXchange Consortium via the PRIDE partner repository with the dataset identifiers PXD061762 and PXD061764.

## Full proteome quantification analysis
Full proteome LFQ intensities were extracted from MaxQuant protein groups files and analyzed with Perseus software V1.6.2.3[66]. Proteins were filtered to remove reverse hits, proteins 'only identified by site' and potential contaminants.

For copy number and concentration estimates, the 'Proteomic Ruler' method was applied as a plugin in Perseus 2.0.11.0[67]. Raw intensities of the six full proteomes of unperturbed cells were filtered to remove reverse hits, proteins 'only identified by site' and potential contaminants, and annotated with average molecular weights from the yeast FASTA sequence database used for MaxQuant processing. The proteomic ruler tool was then applied using average molecular weights to normalize intensities, processing all six data columns separately. An estimate for ploidy was obtained from entries 109469, 108315 and 108196 of the BioNumbers database[68], which yielded an average of 0.022 pg DNA/cell. Assuming 12.1 million base pairs or 0.013 pg per yeast genome, 0.022 pg DNA/cell corresponds to 1.7 genomes/cell. An estimate for intracellular protein concentration was obtained from BioNumbers entries 100427, 100430, 100452, 111978, 109469, 108315, 106225, 100490, which yielded 94 g/l for haploid and 79 g/l for diploid yeast. Interpolation to a ploidy of 1.7 gave an estimated concentration of 83 g/l.

For the full proteome PCA plot (Supplementary Fig. 3a), proteins from all three conditions were analyzed jointly. Proteins were required to include at least three measured datapoints within one condition. Following log transformation, missing data were imputed from a normal distribution with default settings. Data were then subjected to PCA.

For pairwise 'volcano' analyses, proteins were required to include at least three measured datapoints within one condition. Following log transformation, missing data were imputed from a normal distribution with a downshift of 1.8 standard deviations and a width of 0.3 standard deviations. Data were then analyzed with a two-tailed t-test to identify proteins with altered abundance. Non-linear significance cut-offs (i.e., the 'volcano lines') were defined via Perseus' permutation-based FDR calculation (S0 parameter set to 0.1). Proteins with an estimated FDR < 5% were considered to have changed abundance significantly.

For 1D-annotation enrichment, proteins were annotated with GO terms in Perseus and the results of the volcano analyses were analyzed with default settings.

## Generation of compartment marker list
A reference database was created containing 5389 proteins for which abundance estimates in molecules per cell were available[17] (Supplementary Data 2a). Eighteen localization categories were defined: actin-associated, cell wall, COPI coat, COPII coat, cytosol, ER, endosomes, ER-Golgi intermediate compartment (ERGIC), Golgi, lipid droplets, mitochondria, nucleus, nuclear envelope, peroxisomes, plasma membrane, proteasome core, ribosome core, and vacuole. Two additional categories were 'ambiguous' (proteins for which evidence suggested

localization to more than one subcellular compartment) and 'unknown' (proteins for which no assignment was possible). Based on an analysis of the GFP fusion collection[9] (data downloaded from the CYCLoPs web site), the Saccharomyces Genome Database and primary literature, proteins were manually assigned to one of the above categories.

The 2971 protein groups present in steady-state maps of unperturbed yeast were taken from the reference database and used for iterative training of the SVM module in DOM-ABC (see below). The first training set consisted of 868 protein groups, which were the 100 most abundant protein groups in the categories cytosol, ER, mitochondria, nucleus, plasma membrane, and vacuole including endosomes, as well as all available protein groups in the categories Golgi, lipid droplets, nuclear envelope, peroxisomes, proteasome core, and ribosome core. Proteins from the remaining categories were omitted because their localizations are inherently ambiguous. COPI, COPII, and ERGIC proteins show complex distributions at the ER/Golgi interface, and proteins related to the actin cytoskeleton are cytosolic but can associate with the plasma membrane. In addition, cell wall proteins were omitted because the cell wall had to be removed as part of the subcellular fractionation procedure. A first round of training achieved 94% recall, i.e., the SVM prediction agreed with the reference database for 815 protein groups. Furthermore, there were 682 additional protein groups for which a localization was predicted with at least medium confidence and matched the annotation in the reference database. A second training set was assembled by combining the 815 correctly recalled protein groups and the 682 additional protein groups correctly predicted during the first training round. The second round of training achieved 99.9% recall and correctly predicted the localization of 269 additional protein groups, and these results were used to assemble the third training set. This procedure was repeated until, after the seventh iteration, recall of the 1908 protein groups in the training set was 100% and no further protein groups were predicted correctly with at least medium confidence. The resulting final compartment marker set consisted of 1908 protein groups, corresponding to 1937 unique proteins.

## Organellar mapping data analysis in DOM-ABC

The data required to recapitulate the DOM-ABC analyses are provided as Supplementary Data 14. These include trimmed protein groups files (steady-state maps, ER stress maps combined, ER stress maps single) as input, the custom yeast compartment marker list, the Uniprot tab used for gene annotation, the DOM-ABC settings and output.json files, and instructions for processing. Data filtering, annotation, normalization, quality control, and mapping analyses were performed as described[26]. Briefly, the protein groups output file from MaxQuant is loaded into the online tool domaps version 1.0 (accessible at https://domabc.bornerlab.org/QCtool, source code available at https://github.com/JuliaS92/SpatialProteomicsQC/tree/1.0), which formats the data for downstream analysis and quality control. Intensities in each of the six organelle fractions (1000–78,000 x g pellets) are normalized to the total summed intensity across all fractions to obtain 0-to-1 normalized profiles. These can be directly compared between proteins, irrespective of relative abundance, and reflect subcellular distribution. Normalized profiles are then used for all downstream analyses, including calculation of map depth, fraction correlation evaluation, visualization by PCA, protein shift analysis, and compartment classification by SVMs. Relevant parameters were:

(1)  Data annotation. Proteins were annotated with the compartment markers list and current gene names obtained from the Saccharomyces Genome Database.

(2)  Data filtering. Only profiles with at least three consecutive fraction MS intensities and a minimum average MS count of two were retained.

(3)  Principal component analysis (PCA). The 'fix aspect ratio by variability' was deselected to optimize axis scaling for visualization. PC1 versus PC2 were plotted as the most informative principal components to show map resolution. For steady-state maps, all six replicates were combined into one dataset and processed jointly in the 'analysis' module of DOM-ABC. For ER stress maps, data for each condition were separated into three protein groups files and processed individually to generate three.json files. These were then loaded into the benchmark module and jointly subjected to PCA. To generate the PCA plots of individual maps, the joint data were reloaded into the 'analysis' module.

(4)  SVM compartment classification. Default settings were used to train and benchmark SVMs (C parameter range 1-30, gamma parameter range 1–50, 5 iterations with built-in five-fold cross validation). For model training, all markers were used, and the optimized parameters were used for classification (test set proportion set to 0). For performance benchmarking, markers were split 80:20 into training and leave-out test sets (test set proportion set to 0.2). Models were trained with five-fold cross-validation on the training set only. Optimized SVMs were then applied to the test set to evaluate prediction performance via the F1 score (i.e., the harmonic mean of recall and precision). Average F1 scores across 20 sub-samples (i.e., 75%) of the predictions on the test set were then calculated for each organelle. The classes 'lipid droplets' and 'peroxisomes' had too few members to generate informative test sets and were excluded from performance benchmarking. For steady-state maps, all six replicates were combined into one dataset and processed jointly in the 'analysis' module of DOM-ABC to generate a single.json file. This file was then loaded into the 'benchmark' module of DOM-ABC for SVM classification. For ER stress maps, data for each condition were separated into three protein groups files and processed individually in the analysis module to generate three.json files. These were then loaded into the benchmark module and jointly subjected to SVM analysis. The output of the SVM classification is a probability score for the likelihood of model fit for each of the 12 compartments represented in the compartment marker list (see above). For each protein, scores across compartments therefore add up to 1. Protein are assigned to the compartment with the highest score. Assignments are grouped into confidence classes depending on the magnitude of the score: >0.95 = very high, >0.8 = high, >0.65 = medium, >0.4 = low, <0.4 = best guess. For Fig. 1C, predictions with 'very high' and 'high' confidence were further grouped as 'high confidence', and predictions with 'low confidence' and 'best guess' as 'low confidence'.

(5)  Movement-reproducibility (MR) analysis. To detect proteins with localization changes, default settings were applied for data pre-filtering (cosine correlation >0.9) and statistical testing (static data proportion = 0.75, number of iterations = 11). For each protein, the profiles from control cells are subtracted from profiles from DTT- or tunicamycin-treated cells to obtain a 'delta profile'. Proteins without significant changes have delta profiles close to baseline. To identify significantly deviating delta profiles, a robust multivariate outlier test is performed. $P$-values from three replicates are then combined with the Fisher method. The joint p-value is corrected with the Benjamini-Hochberg method and -log10 transformed to obtain the movement (M) score. $M = 2$ means that a protein undergoes a significant movement with an estimated FDR of 1%. Here, an M-score of 1.3 was chosen as cut-off for significance (FDR < 5%). As an additional stringency filter, the direction of movement had to be consistent across replicates. The test therefore calculates

the pairwise Pearson correlation of delta profile replicates (Rep1 vs Rep2, Rep1 vs Rep3 and Rep2 vs Rep3). The median of these three values was chosen as the R-score, with a value of 0.8 as cut-off for reproducibility. Finally, proteins had to have a p-value for movement <0.05 in at least two of the three replicates to qualify as hits. DOM-ABC performs a data quality filtering step prior to MR analysis so that only profiles with high replicate reproducibility (all pairwise cosine correlations >0.9) and only proteins profiled in all three replicates of both compared conditions are included. Hence, the number of proteins in the MR analysis is usually lower than the number of mapped proteins.

### Cytosolic pool analysis

To estimate protein cytosolic pools, a second set of abundance profiles was generated in DOM-ABC that included the six organellar fractions and also the cytosol fraction. Normalized intensities in each fraction were weighted with the corresponding relative protein yields as measured by BCA assay. Weights for each fraction were calculated as average percentual protein recovery across replicates. Weighted intensities were divided by the sum across all fractions to obtain 0-to-1 normalized weighted intensities. These seven datapoint profiles reflect the percentage recovery of a protein across subcellular fractions. The cytosol fraction corresponds to the actual cytosolic pool (which is excluded from the six-datapoint profiles used to generate organellar maps, see above). The sum of the first six fractions indicates a protein's non-cytosolic pool.

For steady-state maps, proteins were annotated with cytosolic pools to identify proteins with potential dual cytosolic and non-cytosolic localizations (Supplementary Data 1c) and determine the extent of organellar leakage during fractionation (Supplementary Fig. 1). For ER stress maps, cytosolic pools were analyzed to identify proteins that shift to or away from the cytosol upon ER stress. First, proteins not quantified in all maps were excluded. Second, proteins minimally had to have a measured cytosolic pool in all three replicates of one treatment condition. For the remaining 2204 proteins, the cytosolic pool change was analyzed in control relative to DTT- or tunicamycin-treated cells with a paired two-tailed t-test. Since the generation of the cytosolic pool data required several processing steps, which may contribute additional noise, the results for DTT- and tunicamycin-treated cells were combined to increase stringency and statistical power. The two individual p-values were combined with the Fisher method, and the joint p-values were corrected for multiple testing with the Benjamini-Hochberg method. To qualify as hits, proteins had to have: an overall FDR < 5%; significant individual p-values with both DTT and tunicamycin treatment ( < 0.05 for the DTT set and <0.1 for the slightly noisier tunicamycin set); an absolute cytosolic pool change >10% with DTT and tunicamycin treatment; the same direction of change (i.e. towards or away from the cytosol) under both conditions. 74 proteins passed these filters (Supplementary Data 5e). Changes in the cytosolic pool are shown only for the DTT data in Fig. 5A, but the identification of significant hits was performed as described here.

### Comparison of SVM localization predictions with reference database

The SVM predictions included 12 annotated subcellular localizations, and the reference database 20. To enable a direct comparison of the localization predictions, the reference database categories 'endosomes' and 'vacuole' were pooled (to just 'vacuole'), and also the categories 'cell wall' and 'plasma membrane' (to just 'plasma membrane'). Proteins with categories not represented in the SVMs were removed ('actin-associated', 'ambiguous', 'COPI', 'COPII', 'ERGIC', 'unknown'). This left 2554 proteins with matching categories. Furthermore, proteins were annotated with the cytosolic pools determined in this study. Proteins with organelle assignments and an

additional cytosolic pool >30% were considered to have a dual localization (Supplementary Data 1c).

### Comparison of SVM localization predictions with previous yeast data

The roughly 900 localization predictions in Nightingale et al.[16] were category-matched to the SVM predictions generated here, where possible, and matched up via protein IDs. Localization predictions for the 775 proteins that could thus be compared across both datasets showed 93% agreement (Supplementary Data 1e).

### Analysis of organellar shifts

To determine formally whether ER stress shifted the fractionation behavior of an organelle, a two-tiered test was performed. For each organelle, the average normalized profile of all compartment marker proteins was calculated, individually for all three map replicates. Average profiles of control and treatment (DTT or tunicamycin) maps were then compared. First, at each of the six profile datapoints, a two-tailed, paired t-test was performed ($n = 3$ replicate profiles). The smallest of the six obtained p-values was multiplied by six to correct for multiple hypothesis testing (Benjamini-Hochberg method). The obtained Q-value was used to determine whether the compared profiles showed a statistically significant difference. Second, the absolute distance of the compared profiles was calculated for each cognate pair of replicates by summing the absolute differences at each of the six datapoints. The average of these three comparisons reflected the magnitude of the profile change. The typical absolute profile distance within replicates was calculated as a reference point. The average inter-replicate scatter across all conditions and organelles was approximately 0.1. Therefore, profile distances <0.1 were considered as negligible, i.e. within the range of typical data noise. Profile distances >0.1 but <0.2 were considered as small, >0.2 but <0.3 as medium, and >0.3 as large. Organelles with a significant shift at FDR < 5% and a profile shift magnitude >0.1 were considered to undergo a systematic shift upon DTT or tunicamycin treatment. Lipid droplets and peroxisomes were excluded from the analysis because they had a too few marker proteins for a robust analysis.

### Identification of proteins shifting towards the ER during ER stress

First, profile Pearson correlation of mapped proteins with the average ER marker profile was determined for control and DTT-treated cells. For each protein, the change of correlation with ER markers was calculated as Delta correlER = (correlationDTT ER) – (correlationCon ER). A positive Delta CorrelER identified proteins that correlated better with ER markers under ER stress. To identify proteins shifting towards the ER, the following filters were applied: hit in the DTT MR analysis (M > 1.3, R > 0.8); positive Delta CorrelER; Pearson correlation with average ER marker profile in DTT-treated cells >0.75; protein has post-ER secretory pathway localization in untreated cells according to reference database or SVM classification. This analysis identified 86 proteins. Inspection of the 410 DTT hits identified three proteins (Ape3, Sln1, Toh1) that narrowly failed the second filter but passed all other filters and had a convincing shift towards the ER in PCA plots. They were therefore added to the final list of 89 proteins (Supplementary Data 7). For three of these (Dcr2, Lcb5, Epo1), UniProt information did not predict transmembrane domains or ER-targeting signal peptides, suggesting that they are peripheral membrane proteins.

### Identification of ER lumenal proteins

Proteins in the reference database (Supplementary Data 2a) were filtered for ER localization. These 315 proteins were annotated with UniProt information on transmembrane domains and ER-targeting signal peptides. The 17 ER proteins with a signal peptide and no transmembrane domains were considered lumenal. Based on the

localization data generated in this study, we manually added to this list two proteins (Fpr2, Fmp27) from the pool of proteins with 'ambiguous' localization in the reference database.

## Identification of post-ER secretory pathway proteins

The 2320 proteins available for MR analysis in DTT-treated versus control cells (Supplementary Data 5b) were annotated with UniProt information on transmembrane domains, ER-targeting signal peptides and mitochondria-targeting transit peptides. Proteins were filtered to contain an ER-targeting signal peptide and/or one or more transmembrane domains but did not contain a mitochondrial-targeting transfer peptide. Remaining proteins were filtered for a predicted localization to post-ER secretory compartments. For this, we principally used the reference database but manually augmented it with data from this study. This procedure identified 243 post-ER secretory pathway proteins present in the DTT profiling set (Supplementary Data 5g).

## Microscopy

Cells were cultured as described above and, where indicated, treated with 8 mM DTT for 2 h or with 2 μg/ml tunicamycin for 3 h. Cells expressing GFP-tagged GPI-anchored proteins under an estradiol-inducible promoter system were cultured in the presence of 50 nM estradiol (Sigma) for 16 h prior and DTT treatment was done in SCD containing 50 nM estradiol and 50 mM HEPES pH 7.5. HEPES was included to avoid quenching of extracellularly exposed GFP by the low pH of SCD medium. Cells from 1 ml culture were harvested by centrifugation at $10{,}000 \times g$ at room temperature for 2 min. The supernatant was removed and cells were resuspended in 20 μl medium. For staining with the silicon-rhodamine HaloTag ligand[69], cells were harvested, resuspended in 30 μl PBS containing 500 nM HaloTag ligand and incubated at 800 rpm at room temperature for 10 min. Three μl cell suspension were mounted on coverslips and covered with 1% w/v agarose pads made with SCD medium. For imaging of GPI-anchored proteins, agarose pads were made with PBS instead of SCD to avoid quenching of GFP fluorescence.

For imaging pulse-chase experiments of Prc1 and Gas3 (Fig. 6d and Supplementary Fig. 8d), cells expressing Prc1-sfGFP or ss-sfGFP-Gas3 under an estradiol-inducible promoter system were grown to early log phase in SCD medium. Cells were either left untreated or treated with 8 mM DTT for 1 h, then treated with 100 nM estradiol for 30 min to induce expression, washed with water to remove the estradiol, resuspended in SCD without or with DTT, and imaged as described above after another 60 min of culture. Thus, untreated cells were exposed to estradiol for 30 min followed by a 1 h chase, and cells synthesizing Prc1-sfGFP and ss-sfGFP-Gas3 during ER stress were treated with DTT for 1 h without estradiol, 30 min with estradiol, and another 1 h without estradiol.

Images were acquired with a Nikon Ti2 widefield microscope equipped with a Nikon Plan Apo 100x/NA 1.45 objective and a Hamamatsu Orca Fusion-BT camera (images for Supplementary Fig. 9c, f) or with a Nikon Ti2-W1 spinning disk confocal microscope equipped with a Yokogama W1 scanhead, a Nikon Plan Apo 100x/NA 1.45 objective and a Zyla 4.2 P CMOS camera (all other images). Images for Fig. 5e and Supplementary Fig. 6e were acquired as Z-stacks with a step size of 200 nm to facilitate subsequent image deconvolution. For the systematic analysis of GFP-tagged nucleoporins (Supplementary Data 8), cells were treated and imaged by widefield microscopy as above. Images were anonymized with the "Blind Analysis Tools" plugin in ImageJ (https://imagej.net/plugins/blind-analysis-tools) before visual analysis to prevent user bias. Puncta formation was assessed independently by two individuals. The resulting rare cases of disagreement were resolved by a second round of joint assessment.

Confocal images for Fig. 5e and Supplementary Fig. 6e were deconvolved with the Richardsson-Lucy algorithm (30 iterations, noise

level set to automatic, background subtraction activated) in NIS Elements software (Nikon). All other confocal images were processed with Fiji software. General background was subtracted in all channels with the rolling ball algorithm (radius = 150 pixels, or 9.75 μm), regions of interest were selected and the display range was adjusted to avoid blank and saturated pixels. Widefield images for Supplementary Fig. 9f were processed the same except that no adjustment was applied to the NLS-mNeon signal.

## Image quantification

Redistribution of ER proteins to the cytosol and ER retention of vacuole and Golgi proteins were quantified from single optical slices of confocal images of untreated, DTT- or tunicamycin-treated cells expressing a sfGFP-tagged protein and the ER marker Sec63-mScarlet. Image analysis and subsequent calculations were done with Fiji software and Python using custom scripts (QuantifyIntensity_ER.ijm and ImageQuantification_Calculations.ipynb; see code availability section). The ER fraction of each sfGFP-tagged protein was determined, i.e. the fraction of ER-localized signal relative to total cell signal. First, a cell border mask was generated by automated global thresholding of the brightfield image. The cell border mask was eroded to eliminate small structures and retain only the cell cortex. Second, a whole cell mask was generated by automated global thresholding of the Sec63 signal and the 'fill holes' command. To identify individual cells, the cell border mask was subtracted from the whole cell mask. Watershed segmentation was applied, single cells were defined by particle analysis (size = 7–29 μm², circularity = 0.5 – 1.0) and stored as regions of interest (ROIs). Cell area and mean pixel intensity were measured within each single-cell ROI. Third, an ER mask was generated based on the Sec63 signal using the Bernsen method for automated local thresholding in a radius of 5 pixels. The single-cell ROIs were overlayed with the ER mask, and single ER particles were defined using particle analysis without size and shape constraints. The resulting separate ER ROIs were combined into one ER ROI per cell and overlayed with the sfGFP signal. ER area and mean pixel intensity were measured within each cell. Fourth, to correct for autofluorescence, mean background pixel intensities for cell and ER signal in the sfGFP channel were determined as above from images of cells expressing Sec63-mScarlet but no green fluorescent protein. The mean background cell and ER pixel intensities were subtracted from the mean cell and ER pixel intensities from sfGFP-containing cells. Last, total ER and cell signal were calculated by multiplying the respective areas by the background-subtracted mean pixel intensities, and the ER fraction was calculated by dividing the total ER signal by the total cell signal.

ER retention of plasma membrane proteins was quantified from single optical slices of confocal images of untreated or DTT-treated cells expressing a sfGFP-tagged protein, Sec63-mScarlet and the nuclear marker Pus1-Halo. The peripheral ER and the plasma membrane cannot be resolved by light microscopy, so that ER retention of plasma membrane proteins was quantified by determining the nuclear envelope fraction, i.e. the fraction of nuclear envelope-localized signal relative to total cell signal. The cell border mask and the whole cell mask were generated as above, except that the Gas1 and Gas3 signal, respectively, instead of the Sec63 signal were used for the whole cell mask. Cell area, mean pixel intensity and an ER mask based on the Sec63 signal were determined as above. An additional nuclear mask was generated using automated global thresholding of the Pus1 signal. The ER mask and the nuclear mask were overlayed and only the nuclear envelope was kept. Nuclear envelope area and mean pixel intensity were measured within each cell, and the remainder of the analysis was done as above.

Cytosolic clusters of the nucleoporins were quantified from widefield images of untreated and DTT-treated strains expressing an mNeonGreen-tagged nucleoporin, the ER marker mCherry-Ubc6 and Pus1-Halo. Images were acquired as Z-stacks with a step size of 200 nm

and were deconvolved as above. A maximum intensity projection of four optical slices around the cell mid-section was generated and used for all subsequent steps. Cytosolic nucleoporin clusters were quantified with custom scripts (Puncta_counter.ijm and Nuppuncta_Quantification.ipynb; see code availability section). The single cell mask and cell ROIs were generated as described above. Two further masks were generated: a nuclear mask based on the Pus1 signal and an initial puncta mask based on the 1% brightest pixels of the nucleoporin signal. Individual puncta were defined using particle analysis (size = $0.01 - 0.2\ \mu m^2$, no circularity constraints), yielding a refined puncta mask. To eliminate nucleoporin signal at the nuclear envelope, the nucleus mask was subtracted from the refined puncta mask, yielding the final cytosolic puncta mask. The cytosolic puncta mask was applied to the nucleoporin channel, and puncta were analyzed in each single-cell ROI using particle analysis with the same parameters as above. The number of puncta for each cell was determined and the average number of cytosolic puncta per cell was calculated.

### Western blotting
Western blotting was done as described[35] with primary mouse antibodies against GFP (clones 7.1/13.1 from Roche, AB_390913), mNeonGreen (clone 32F6 from Chromotek, AB_2827566), mCherry (1C51 from Abcam, AB_3242120) and anti-Pgk1 (clone 22C5D8 from Abcam, AB_10861977) and fluorescent secondary antibodies (goat anti-mouse Alexa-680 from Invitrogen, AB_141436). Fluorescence was detected with an Odyssey CLx imaging system (LI-COR).

### Nuclear import assay
Yeast (SSY4574) were cultured as described above. Control cells were diluted to $OD_{600} = 0.15$ and grown for 1 h. Cells to be exposed to ER stress were diluted to $OD_{600} = 0.25$, treated with 8 mM DTT and grown for 1 h. Each culture was split in two, one new culture received no further treatment and was used to determine background auto-fluorescence in the GFP channel, the other new culture was treated with 500 nM estradiol for 50 min to induce expression of NLS-mNeonGreen. Cells were imaged by widefield microscopy as described above and single optical sections in z were acquired.

Image quantification was done with Fiji software using a custom script (QuantifyNucImport.ijm and NucImportQuantification_Calculations.ipynb; see code availability section). Background signal was subtracted in all channels with the rolling ball algorithm as above and a cell mask was generated based on the combined BFP and brightfield images using global automatic pixel intensity thresholding followed by watershed segmentation. A nucleus mask was generated the same way based on the Pus1-mScarlet image. Cell and nucleus masks were used to measure the areas, mean BFP and mean mNeonGreen pixel intensities of individual cells and nuclei. Cell measurements were done for objects with a size of $9-27\ \mu m^2$ and a circularity of 0.5–1, nucleus measurements were done for objects with a size of $1.1-7\ \mu m^2$ and a circularity of 0.1–1. Nucleus measurements were assigned to the corresponding cell measurements. Cell measurements with no or more than one nucleus measurement were removed. To determine nuclear enrichment of NLS-mNeonGreen, mean cell and nuclear mNeonGreen pixel intensities were first determined from cells not treated with estradiol (>250 cells per condition and replicate). These background fluorescence values were substracted from the mean cell and nuclear pixel intensities in estradiol-treated cells, which expressed NLS-mNeonGreen. Cells with background-corrected mean pixel intensities of less than 1 were removed. Nuclear enrichment in the remaining cells was then calculated by dividing the background-corrected mean pixel intensity of each nucleus by the background-corrected mean pixel intensity of the corresponding cell (>200 cells per condition and replicate). Autofluorescence in the BFP channel was negligible, so that nuclear enrichment of BFP was calculated without background correction by dividing the mean pixel intensity of each nucleus by the

mean pixel intensity of the corresponding cell ( > 250 cells per condition and replicate). Data from three biological replicates were combined, yielding >800 cells analyzed per condition.

### Statistics and reproducibility
**Sample size.** No statistical methods were used to predetermine sample size. The number of replicates obtained for dynamic organellar mapping ($n = 3$ biological replicates of technical duplicates for steady state maps; $n = 3$ biological replicates for ER stress maps) was chosen based on previous experience with the reproducibility of the technique.

**Data exclusions.** No data were excluded except during image quantification for Supplementary Figs. 6b, 7c–e, 8d and f. There, a small number of cells with an ER or nuclear envelope fraction >1 was filtered out during automated image analysis. Such fractions >1 are impossible and reflect inaccurate masking of the cell area.

**Replication.** The proteomics experiments were done in biological triplicate of technical duplicates (steady state maps) or in biological triplicate (ER stress maps). Microscopy experiments shown in Figs. 5c, e, 6c, d, 7b, d, 8b, e and Supplementary Figs. 6c, e, 7a, b, 8a–c, e, 9a, e and g were done at least twice to test for reproducibility of the observed localizations. The nuclear import assay shown in Fig. 8f and Supplementary Fig. 9f was done in biological triplicate. Western blots shown in Fig. 6e and Supplementary Figs. 6a, d and 9a were done once. Image quantification shown in Supplementary Figs. 6b, 7c–e, 8d and f was done once and cells were quantified from three different fields of view each.

**Randomization.** The experiments consisted of comparisons of genetically different or differently treated yeast strains, so no randomization was necessary.

**Blinding.** Blinding was not relevant except for the data underlying Fig. 8c, in which two individuals independently assessed microscopy images to determine whether various nucleoporins formed cytosolic puncta. For these assessments, the "Blind Analysis Tool" plugin was used to blind the evaluators to the identity of the imaged strains (see above).

### Reporting summary
Further information on research design is available in the Nature Portfolio Reporting Summary linked to this article.

## Data availability
The proteomic datasets generated during this study are available from the ProteomeXchange Consortium via the PRIDE partner repository with the dataset identifiers PXD061762 and PXD061764. Source data are provided with this paper.

## Code availability
The web application DOM-ABC[26] is available at https://domabc.bornerlab.org, the source code at https://github.com/JuliaS92/SpatialProteomicsQC, and version 1.0.0 at https://doi.org/10.5281/zenodo.8219481. Custom scripts for image quantification are available at https://gihub.com/SchuckLab/Image_analysis_tools_Platzek and at https://doi.org/10.5281/zenodo.17508539.

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

## Acknowledgements

We thank Alexandra Davies for help with mass spectrometry, Kai Johnsson for the SiR HaloTag ligand, the Nikon Imaging Center at Heidelberg University for assistance, and Anne Schlaitz and all Schookees for incisive comments on the manuscript. This work was supported by grants SCHU 2364/3-1 and SFB-1638/1 - P02 from the German Research Foundation (DFG) to SS. GHHB would like to thank Matthias Mann for his continued support. The authors gratefully acknowledge the data storage service SDS@hd supported by the Ministry of Science, Research and the Arts Baden-Württemberg (MWK) and the DFG through grant INST 35/1503-1 FUGG. For the publication fee the authors acknowledge financial support by Heidelberg University. The authors used ChatGPT (OpenAI, GPT-4, July 30, 2024) for assistance with Fiji and Python scripting.

## Author contributions

G.B., A.P., and S.S. conceptualized the experiments. A.P., K.O., and J.S. carried out the experimental studies. G.B. performed the statistical analysis of the Dynamic Organellar Maps data. G.B. and S.S. supervised the work and wrote the manuscript with input from all authors.

## Funding

## Competing interests

The authors declare no competing interests.
