## [Transparent Peer Review file · Nature Communications]

Dynamic Organellar Mapping in yeast reveals extensive protein localization changes during ER stress

Corresponding Author: Dr Sebastian Schuck

Version 0:

Reviewer comments:

Reviewer #1

(Remarks to the Author)

Platzek et al present a subcellular map of the *S. cerevisiae* proteome, building on a the 'dynamic organellar maps' workflow established by the authors in previous publications for mammalian cells.

The authors then characterize subcellular proteome dynamics in cells exposed to two agents that perturb the ER: DTT and tunicamycin. They find overall very similar responses to those two drugs on the proteome (with the exception of the effect on mitochondria).

The experimental design and execution is technically very robust. The paper is well-written. Datasets are being made available via supplemental excel tables that are reasonably straightforward to browse even for a non-technically versed audience.

I only have a few comments that should be addressed in writing and with minor new computational analyses before submission of a revised version:

1) Completeness of the dataset: There is a delta of ~1,000 proteins that seem quantified in the proteome but not localized. A further ~1,500 proteins (of assumed 5,400 expressed proteins total) are not detected.

What are the technical limitations that explain these deltas, aside from low protein abundance? Are any subcellular locations particularly likely to not be localizable based on the DOM workflow, when comparing with microscopy-based datasets?

2) Side effects of spheroplast generation:

Can the authors comment on the limitations induced by zymolyase treatment? Are cell wall or plasma membrane proteins expected to be faithfully localizable?

3) There are conceptually two types of 'movement' of proteins, which unfortunately are hard to distinguish in a systematic fashion: True relocalization of individual proteins in or out of organelles, and overall changes in the fractionation behavior due to altered morphology of an entire organelle (as evidenced by the global ER proteome changes observed by the authors). I think this dichotomy deserves a more formalized distinction, both in how it is discussed in the description of the results, and in the data analysis pipeline.

Something along the lines of: a) for all consensus organelles, test whether a majority remain unchanged upon perturbation or whether they change in unison. b) in case of the latter, attempt to correct the behavior of individual proteins by an average behavior of the majority so that outliers can still be identified correctly.

Minor:

One of the classic, tagging and microscopy-based studies should be cited: Kumar et al, 2002 DOI 10.1101/gad.970902.

Reviewer #2

(Remarks to the Author)

In this study, Platzek et al establish dynamic organellar mapping in *S. cerevisiae* and apply it to analyze the cellular response to ER stress. This systematic analysis provides an unbiased picture of ER stress-induced changes in protein

localization, significantly broadening our view and understanding of the many consequences of ER stress on cell physiology. Whilst revealing previously unnoticed ER stress-induced protein movements, this work recovers and extends the characterization of previously known processes such as protein reflux or ER retention. Overall, this study provides a valuable database that not only greatly expands the current knowledge of ER stress-induced processes -opening up new avenues of investigation- but also demonstrates the power and feasibility of DOM in yeast.

1. It appears that the fractionation procedure does not equally preserve the organelle compartments equally. This is illustrated in Figure S1 where the “contamination” of the cytoplasmic fraction by vacuolar proteins is more important than its contamination by ER or mitochondrial proteins. This figure lacks a panel showing nuclear proteins. According to Table S1, the number of nuclear proteins with a significant cytoplasmic pool is increased. While this could reflect dual localization, as suggested by the authors, it seems unlikely that this is the case for all of these proteins. For example, proteins such as the Pol II subunit Rpb2, which has a cytoplasmic fraction of 0,79 or Nup2 (cytoplasmic fraction 0,73), would not be expected to have such elevated cytoplasmic pools. It is not clear to me how these proteins can still have a nuclear SVM prediction (Table S1C) despite being labeled with “cytoplasmic pool>30%”. Could the authors clarify this aspect and discuss how this might affect the observed remodeling of the nuclear proteome?

2. While it is not surprising that Tm and DTT induce some specific protein relocation events, given their different mode of action, one would expect ER stress-specific events to be shared by both inducers, as is the case for ER reflux. However, the relocation of reticulon homology domain proteins or that of Nups, for example, appears to be DTT-specific. Is this truly the case? The authors argue (line 194) that DTT is known to “trigger a more extensive remodeling of ER morphology” but this is using a rather high DTT concentration (8mM) compared to the classical 1 or 2mM DTT treatment used in UPR studies. Could the effect of Tm-induced ER stress be re-evaluated on some of these localization changes (e.g., Nups and reticulon-like proteins) using Tm concentrations that lead to a level of ER stress comparable to that induced by 8 mM DTT treatment (estimated by the level of UPR activation and the reduction of ER redox status)? Although DTT-specific effects would remain interesting, these controls are important to assert for ER stress-mediated events.

3. Minor points:

- Figure 1B: on the PCA plots, color coding makes it difficult to distinguish the “nucleus”, “undefined” and “unknown” compartments.
- Figure 5C and S6A: the microscopy pictures used to validate ER reflux are not convincing, particularly when compared to the DOM data (Figure 5B) or to the pictures from Lajoie et al 2020. Without a quantification or the imaging of different reflux substrates, these images do not really support the DOM data.
- Figure 5D: could the cytoplasm be highlighted?
- Table S4: the header in both A (DTT) and B (Tm) says tunicamycin

Reviewer #3

(Remarks to the Author)

This manuscript, "Dynamic Organellar Mapping in yeast reveals extensive protein localization changes during ER stress," presents a technically solid and clearly written study. The authors use Dynamic Organellar Maps (DOMs) in *Saccharomyces cerevisiae* to detect proteome-wide changes in protein localization under ER stress, using DTT and tunicamycin. The data are supported by proteomic analysis and some microscopy validation, and the authors also provide a web-based tool to explore the dataset.

Overall, this is a well-executed and useful dataset that will interest researchers in yeast biology and proteomics. The design and analysis are performed with care, and the conclusions are mostly supported by the data. The user-friendly interface for the data is nice and useful (although require more documentation).

However, I believe that this study does not bring the level of conceptual novelty or depth expected for publication in *Nature Communications*. The DOM method and software were already developed and published by the same group. Similar spatial proteomics work was also done in yeast before. In this study, many of the findings are interesting but remain descriptive and without deeper mechanistic explanation. Below are the specific points that I suggest the authors to address.

Major Comments

1. Availability of Proteomics Data

The Raw MS data is not available for examination. It is very important that the raw and processed proteomics data are deposited in PRIDE/ProteomeXchange.

2. ER Reflux and Membrane Integrity

The authors conclude that the ER membrane stays intact during fractionation because only a few ER luminal proteins appear in the cytosol. This supports selective reflux, but the conclusion would be stronger with more controls — for example, protease protection assays or an inert luminal marker like Kar2ss-GFP. Also, Figure 5C should be quantified to show how many cells display cytosolic fluorescence, and how the signal changes in distribution.

3. Nucleoporin Relocalization (Figure 8)

The observation that nucleoporins relocalize during ER stress is very interesting, but this section can be improved:

It would be good to confirm by Western blot or fractionation that full-length nucleoporins are detected outside the nuclear envelope.

Ubc6 is used as an ER marker in Figure 8B. Other markers for nuclear envelope or vesicles should be tested as well, and other nucleoporins should be analyzed.

The authors should discuss if the puncta could be condensates, stress granules, or compartments for protein quality control, and not only products of NPC disassembly.

Quantitative analysis of relocalization (e.g., how frequent it is among cells, or how fast it happens) would help.

Since all conclusions are based on fluorescent fusions, Western blots should be used to confirm that the proteins remain full-length under stress and are not degraded or cleaved (see also below).

4. Fusion Protein Integrity

Please add a Western blot for selected tagged proteins (both control and stress). This would be useful to confirm that the fluorescence corresponds to full-length fusion proteins. This is especially important for membrane proteins that may be cleaved under stress.

5. Membrane Contact Site Proteins

It would be interesting to look at known membrane contact site proteins, such as those described by Castro et al. (eLife, 2022). These proteins often link organelles and may not be well resolved in DOM PCA plots. Their behavior under ER stress could show changes in inter-organelle organization.

6. Quantitative Microscopy Analysis

The microscopy images are helpful, but the analysis is mostly qualitative. Please add quantitative data — such as percentage of cells showing relocalization, or signal ratio between organelle and cytosol. This would help show the strength and consistency of the results.

7. Abundance Changes in Relocalized Proteins

The analysis is mostly based on normalized localization profiles and does not include much about protein abundance. The authors say that most relocalized proteins do not change abundance, but actually 40% of them do in DTT, and 28% in tunicamycin. This is not a small fraction. It is important to demonstrate and discuss how abundance and localization changes are connected, and whether they might reflect coordinated responses.

8. Use of Tunicamycin

Although tunicamycin is included in the study, it is not discussed much. The authors should explain more clearly why it was used, and whether the localization changes it causes were confirmed or validated. It is unclear how tunicamycin-specific effects compare to DTT.

9. Limitations of PCA

Many results are shown using PCA plots. PCA is good for visualization but may oversimplify complex data. The authors should discuss this limitation and explain if they validated the PCA results in another way.

Minor Comments

1. ER Membrane Remodeling and Terminology

Rtn1, Yop1, and Dpm1 relocalize into stress-induced puncta, and their sedimentation behavior changes. This suggests major remodeling of the ER membrane. The authors should make it clear that “membrane integrity” here means that the luminal content is retained, but does not exclude formation of subdomains or vesicles.

2. Redundancy in Figure 8A and 8C

Both panels show PCA plots of nucleoporins. It may be better to combine them, for example by using color to show which ones form puncta in Figure 8A.

3. Mitochondrial Shifts in Figure 3A

Mitochondrial proteins show some shift, especially with tunicamycin, but this is not mentioned in the text. A short note on this would be helpful.

4. Interactive Tools Documentation

The interactive tools provided for data exploration are very nice and helpful. However, the documentation could be improved by including more detailed explanations of the different output types and what they represent. This would make the tools easier to understand and more accessible for users who are less familiar with DOM analysis.

Reviewer #4

(Remarks to the Author)

Version 1:

Reviewer comments:

Reviewer #1

(Remarks to the Author)

The authors have sufficiently addressed my concerns, with one small exception:

In response to my concern about distinguishing between individual protein vs global organelle changes (major point 3) they have devised a convincing test for global organelle behavior change. They do, however, note that correction for such global effects are non-trivial and need to be addressed in future work. It seems appropriate to include this limitation in the main text. Here is a suggestion:

(line 253):

...reflect shifts distinct from the overall organelle behavior.

In case of the ER, 90% of constituent proteins showed a consistent shift, allowing us to attempt to correct for this effect globally and detect individual protein shifts with greater sensitivity. We therefore subtracted...

(line 259, insert)

For other compartments with less uniform profiles of their constituent proteins, global corrections are less straightforward.

(Remarks on code availability)

Reviewer #2

(Remarks to the Author)

The authors have addressed all my comments with clear and useful explanations or appropriate additional experiments. Congratulations on generating this data set with clear “Interactive Tools” documentation, which will be precious in guiding the readers through DOM analysis.

(Remarks on code availability)

Reviewer #3

(Remarks to the Author)

I appreciate the authors' efforts in addressing my comments; although I am not fully convinced on all points, I have no further requests.

(Remarks on code availability)

I reviewed the additions the authors added to the supplement Excel files. The authors additions allow better understanding of the output (as was suggested)

Point-by-point response

Reviewer #1

Platzek et al present a subcellular map of the *S. cerevisiae* proteome, building on the 'dynamic organellar maps' workflow established by the authors in previous publications for mammalian cells. The authors then characterize subcellular proteome dynamics in cells exposed to two agents that perturb the ER: DTT and tunicamycin. They find overall very similar responses to those two drugs on the proteome (with the exception of the effect on mitochondria).

The experimental design and execution is technically very robust. The paper is well-written. Datasets are being made available via supplemental excel tables that are reasonably straightforward to browse even for a non-technically versed audience. I only have a few comments that should be addressed in writing and with minor new computational analyses before submission of a revised version:

1. Completeness of the dataset: There is a delta of ~1,000 proteins that seem quantified in the proteome but not localized. A further ~1,500 proteins (of assumed 5,400 expressed proteins total) are not detected. What are the technical limitations that explain these deltas, aside from low protein abundance? Are any subcellular locations particularly likely to not be localizable based on the DOM workflow, when comparing with microscopy-based datasets?

We thank the reviewer for these helpful questions. We have now compared the abundance of (1) the 5400 proteins contained in our reference database, (2) the ≈ 1500 proteins not quantified in whole cell lysates, (3) the ≈ 900 proteins quantified in whole cells lysates but not profiled, and (4) the ≈ 3000 proteins that were quantified, profiled and assigned to a subcellular location. The 1500 proteins we could not quantify were biased towards low abundance and were, on average, in the 25th abundance percentile of the 5400 proteins in the reference database. Among them were also some abundant proteins we could not quantify, including small transmembrane proteins (e.g. Pmp1, Pmp2), which are difficult to analyze by mass spectrometry, and cell wall proteins (e.g. Cwp2, Nfg1, Flo1), which were presumably lost during cell wall removal as part of the cell lysis procedure. The 900 proteins we could quantify but not profile were also typically of low abundance and were, on average, in the 29th abundance percentile of the 3900 quantified proteins. A protein was only profiled if it could be found in at least three consecutive subcellular fractions and in all six replicate maps. Hence, the main factor that caused proteins to go unquantified or unprofiled appears to be their low abundance.

Furthermore, we have compared the compartment distributions of all 5400 expressed proteins and the 3000 profiled proteins. These distributions are almost identical, indicating that no subcellular location is particularly challenging to profile and localize. We have added explanations of these issues (lines 89-92 and 110-114) and included a plot of the compartment distributions as a new Figure S2B.

2. Side effects of spheroplast generation: Can the authors comment on the limitations induced by zymolyase treatment? Are cell wall or plasma membrane proteins expected to be faithfully localizable?

Zymolyase treatment may indeed cause cleavage and removal of plasma membrane and cell wall proteins. As a result, some plasma membrane and especially cell wall proteins may have been lost from the analysis (see above). For this reason, we did not maintain 'cell wall' as an independent localization classifier but merged it with the classifier 'plasma membrane'. However, plasma membrane and cell wall proteins are not significantly depleted among the localized proteins relative to all proteins in the reference database (new Figure S2B). This observation argues that they are, overall, still faithfully localizable.

3. There are conceptually two types of 'movement' of proteins, which unfortunately are hard to distinguish in a systematic fashion: True relocalization of individual proteins in or out of organelles, and overall changes in the fractionation behavior due to altered morphology of an entire organelle (as evidenced by the global ER proteome changes observed by the authors). I think this dichotomy deserves a more formalized distinction, both in how it is discussed in the description of the results, and in the data analysis pipeline. Something along the lines of: a) for all consensus organelles, test whether a majority remain unchanged upon perturbation or whether they change in unison. b) in case of the latter, attempt to correct the behavior of individual proteins by an average behavior of the majority so that outliers can still be identified correctly.

It would indeed be very beneficial to be able to disentangle overall organelle shifts and individual protein shifts in the manner outlined by the reviewer. We have implemented the first step of such an analysis and tested for each major organelle whether there is an overall shift. We have added this analysis as a new Table S4A, describe how it was done in lines 845-865 and present the results in lines 208-212. These additions nicely complement the visual impression conveyed by the PCA plots (Figure 3A). Furthermore, we have revised the text to more clearly distinguish between the two types of movement (lines 252-254).

To then correct the behavior of individual proteins by an average behavior of the organelle majority, as envisaged by the reviewer, raises complex data analysis issues we cannot solve at this point. This correction would work for an organelle like the ER, for which >90% of its constituent proteins show a similar shift. However, it is not easily possible for other organelles. For instance, the Golgi shows a significant shift overall but only 40% of its constituent proteins shift upon DTT treatment as defined by the MR analysis; the remaining 60% do not. Hence, there is no consistent average behavior that could be used for correction. In the future, we aim to devise a local outlier test to ask if the neighborhood of a protein changes. Here, we have solved this problem 'manually' by evaluating every protein shift individually, based on a joint interpretation of all available data.

Minor points:

4. One of the classic, tagging and microscopy-based studies should be cited: Kumar et al, 2002 DOI 10.1101/gad.970902.

We now cite this study in the introduction and the discussion (lines 22 and 463).

Reviewer #2

In this study, Platzek et al establish dynamic organellar mapping in *S. cerevisiae* and apply it to analyze the cellular response to ER stress. This systematic analysis provides an unbiased picture of ER stress-induced changes in protein localization, significantly broadening our view and understanding of the many consequences of ER stress on cell physiology. Whilst revealing previously unnoticed ER stress-induced protein movements, this work recovers and extends the characterization of previously known processes such as protein reflux or ER retention. Overall, this study provides a valuable database that not only greatly expands the current knowledge of ER stress-induced processes -opening up new avenues of investigation- but also demonstrates the power and feasibility of DOM in yeast.

1. It appears that the fractionation procedure does not equally preserve the organelle compartments. This is illustrated in Figure S1 where the “contamination” of the cytoplasmic fraction by vacuolar proteins is more important than its contamination by ER or mitochondrial proteins. This figure lacks a panel showing nuclear proteins. According to Table S1, the number of nuclear proteins with a significant cytoplasmic pool is increased. While this could reflect dual localization, as suggested by the authors, it seems unlikely that this is the case for all of these proteins. For example, proteins such as the Pol II subunit Rpb2, which has a cytoplasmic fraction of 0,79 or Nup2 (cytoplasmic fraction 0,73), would not be expected to have such elevated cytoplasmic pools. It is not clear to me how these proteins can still have a nuclear SVM prediction (Table S1C) despite being labeled with “cytoplasmic pool >30%”. Could the authors clarify this aspect and discuss how this might affect the observed remodeling of the nuclear proteome?

Organelles are indeed not equally preserved during subcellular fractionation. Unlike most ER and mitochondrial proteins, nuclear proteins typically have large cytosolic pools. We think this is due to disruption of the nuclear envelope during cell lysis and homogenization. Yeast has no nuclear lamins, so that yeast nuclei are less mechanically stable than mammalian nuclei. As a result, soluble nuclear proteins are liberated and behave similarly to cytosolic proteins, i.e. are mostly found in the 78k supernatant. By contrast, nuclear envelope proteins behave similarly to ER proteins (see Figure 1B and the simplified PCA plot below). Although many nuclei are presumably broken, soluble nuclear proteins can still be distinguished from cytosolic proteins. This is because their non-cytosolic pools (on average 53% versus 27% for genuine cytosolic proteins) show a characteristic distribution across the six organelle fractions, which are visualized in the PCA plots (see Average Protein Profiles in PCA Profile tab of Supplemental Database 2 and line plot below). This outcome shows the strength of organellar mapping: as long as the constituent proteins of an organelle share a characteristic behavior during subcellular fractionation, they can be recognized as belonging to the same organelle even if that organelle is severely damaged and fragmented. We have added a note on the behavior of nuclear proteins in the results section (lines 97-99) and included a plot of the cytosolic pool estimates for nuclear proteins as a new Figure S1D.

Left: Simplified PCA plot illustrating segregation of nuclear and nuclear envelope proteins. Right: Average abundance profiles of cytosolic and nuclear marker proteins at steady state, showing that their non-cytosolic pools have distinct distributions. Correlation with these average profiles of organelle marker proteins makes it possible to distinguish cytosolic and nuclear proteins.

2. While it is not surprising that Tm and DTT induce some specific protein relocation events, given their different mode of action, one would expect ER stress-specific events to be shared by both inducers, as is the case for ER reflux. However, the relocation of reticulon homology domain proteins or that of Nups, for example, appears to be DTT-specific. Is this truly the case? The authors argue (line 194) that DTT is known to “trigger a more extensive remodeling of ER morphology” but this is using a rather high DTT concentration (8 mM) compared to the classical 1 or 2 mM DTT treatment used in UPR studies. Could the effect of Tm-induced ER stress be re-evaluated on some of these localization changes (e.g., Nups and reticulon-like proteins) using Tm concentrations that lead to a level of ER stress comparable to that induced by 8 mM DTT treatment (estimated by the level of UPR activation and the reduction of ER redox status)? Although DTT-specific effects would remain interesting, these controls are important to assert for ER stress-mediated events.

We thank the reviewer for prompting us to evaluate the similarities between DTT and tunicamycin more thoroughly. The organellar mapping data indicate that the protein localization changes induced by tunicamycin and DTT are qualitatively similar but typically more pronounced with DTT. To test this trend experimentally, we have asked whether tunicamycin also triggers formation of Rtn1 puncta, redistribution of Prc1 and Gas1 to the ER, and clustering of Nsp1, Nup116 and Kap95 in the cytosol. In all cases, tunicamycin caused the same localization changes as DTT, although the effects were typically weaker. Thus, we believe that localization changes are generally shared by both stressors and we have added this information as new Figures S6E, S7B, S8B, S9A and S9E and describe it in lines 310, 357-358, 377-379 and 432-433.

Regarding ER stressor concentrations, we agree that titrating DTT and tunicamycin to elicit similar levels of ER stress becomes important when the two stressors initially cause qualitatively different behaviors. We have not done this here because we wanted to use DTT and tunicamycin at concentrations that achieve the maximum ER stress possible with each drug. Furthermore, 8 mM has been our standard DTT concentration for some time (starting with Schuck et al, JCB 2009).

Minor points:

3. Figure 1B: on the PCA plots, color coding makes it difficult to distinguish the “nucleus”, “undefined” and “unknown” compartments.

This issue is part of the general problem that only a limited number of easily distinguishable colors is available to highlight compartments in crowded PCA plots. We provide a solution in the ‘PCA plots’ tab in Supplemental Database 2, in which each compartment can be individually selected or deselected so that the reader can focus on the compartments of interest. We have added notes to the legends of Figures 1B and 3A to direct the reader to Supplemental Database 2 (lines 1253-1254 and 1277-1278).

4. Figure 5C and S6A: the microscopy pictures used to validate ER reflux are not convincing, particularly when compared to the DOM data (Figure 5B) or to the pictures from Lajoie et al 2020. Without a quantification or the imaging of different reflux substrates, these images do not really support the DOM data.

We have now quantified microscopy images to determine the ER fraction of Sil1-sfGFP-HDEL and Ero1-sfGFP, i.e. the fraction of their signal that overlaps with the signal of the ER transmembrane protein Sec63. This quantification shows that the ER fraction of Sil1-sfGFP-HDEL decreases upon DTT treatment whereas the ER fraction of Ero1-sfGFP stays essentially the same (new Figure S6B). This result agrees with the interpretation that Sil1 but not Ero1 relocates to the cytosol.

Nevertheless, as pointed out by the reviewer, the stress-induced redistribution of Sil1-sfGFP-HDEL to the cytosol is subtle compared with the proteomic data (which show a DTT-induced increase in the cytosolic pool of endogenous Sil1 from almost zero to 50%) and also compared to the images in Lajoie et al, 2020. Importantly, Lajoie and colleagues detected ER reflux for artificial model proteins consisting of an ER signal sequence, a fluorescent protein and an ER retention signal. In addition, they observed an inverse relationship between the size of their model proteins and the efficiency of ER reflux. Hence, their small model proteins of about 30 kDa may redistribute to the cytosol more efficiently than Sil1-sfGFP-HDEL, which is >70 kDa in size. For the same reason, it is plausible that the endogenous Sil1 detected by mass spectrometry, which is <50 kDa in size, redistributes more efficiently than Sil1-sfGFP-HDEL.

Overall, we think our proteomic data provide solid evidence that a specific subset of luminal ER proteins redistributes to the cytosol. ER reflux is one potential mechanism to mediate this redistribution, although this notion needs to be tested for each individual protein. To avoid confusion, we have revised the text to clarify that our data identified candidates for ER reflux cargos, not a definitive list (lines 279-281, 295-297 and 453).

5. Figure 5D: could the cytoplasm be highlighted?

We have highlighted the cytosol cluster, as requested. It is important to bear in mind that the key criterion for identifying Rtn1, Yop1 and Dpm1 as proteins moving to the cytosol is their increased cytosolic pool, i.e. the fraction of the proteins in the 78k supernatant (Figure 5A). We have added a note to the revised legend of Figure 5D to emphasize this point (lines 1305-1307).

6. Table S4: the header in both A (DTT) and B (Tm) says tunicamycin

Thanks for pointing out this mistake, which we have corrected.

Reviewer #3

This manuscript, "Dynamic Organellar Mapping in yeast reveals extensive protein localization changes during ER stress," presents a technically solid and clearly written study. The authors use Dynamic Organellar Maps (DOMs) in *Saccharomyces cerevisiae* to detect proteome-wide changes in protein localization under ER stress, using DTT and tunicamycin. The data are supported by proteomic analysis and some microscopy validation, and the authors also provide a web-based tool to explore the dataset. Overall, this is a well-executed and useful dataset that will interest researchers in yeast biology and proteomics. The design and analysis are performed with care, and the conclusions are mostly supported by the data. The user-friendly interface for the data is nice and useful (although require more documentation).

However, I believe that this study does not bring the level of conceptual novelty or depth expected for publication in *Nature Communications*. The DOM method and software were already developed and published by the same group. Similar spatial proteomics work was also done in yeast before.

The novelty of our work is provided by two key advances. First, we apply organellar mapping in a comparative setting to systematically uncover protein relocalization events. Second, we obtain a vastly expanded picture of proteome remodeling during ER stress. By contrast, the only previous spatial proteomics study in yeast (contained within the review article Nightingale et al., *Curr Opin Chem Biol*, 2019) was limited to the steady-state localization of less than 20% of all yeast proteins.

In this study, many of the findings are interesting but remain descriptive and without deeper mechanistic explanation. Below are the specific points that I suggest the authors to address.

1. Availability of Proteomics Data. The Raw MS data is not available for examination. It is very important that the raw and processed proteomics data are deposited in PRIDE/ProteomeXchange.

The raw MS data have been deposited to the ProteomeXchange Consortium via the PRIDE partner repository with the dataset identifiers PXD061762 and PXD061764 and are already available for review (access information was provided along with the submitted manuscript). We will make the datasets publicly available as soon as the manuscript has received a DOI.

2. ER Reflux and Membrane Integrity. The authors conclude that the ER membrane stays intact during fractionation because only a few ER luminal proteins appear in the cytosol. This supports selective reflux, but the conclusion would be stronger with more controls — for example, protease protection assays or an inert luminal marker like Kar2ss-GFP. Also, Figure 5C should be quantified to show how many cells display cytosolic fluorescence, and how the signal changes in distribution.

We have now quantified microscopy images to determine the ER fraction of Sil1-sfGFP-HDEL and Ero1-sfGFP, i.e. the fraction of their signal that overlaps with the signal of the ER transmembrane protein Sec63. This quantification shows that the ER fraction of Sil1-sfGFP-HDEL decreases upon DTT treatment whereas the ER fraction of Ero1-sfGFP stays essentially the same (new Figure S6B). The observed decrease in the ER fraction of Sil1-sfGFP-HDEL is subtle compared with the strong increase in the cytosolic pool of endogenous Sil1 observed by mass spectrometry. This difference

between endogenous Sil1 (<50 kDa) and Sil1-sfGFP-HDEL (>70 kDa) may be explained by the fact that the efficiency of ER reflux is inversely proportional to the size of cargo proteins (Lajoie et al, 2020; also see response to minor point 4 of reviewer 2). Thus, the image quantification supports the interpretation that Sil1 but not Ero1 relocates to the cytosol.

Further work is needed to determine to what extent the observed cytosolic redistribution of each redistributing luminal ER proteins is mediated by ER reflux. To avoid confusion, we have revised the text to clarify that our data show cytosolic redistribution of a subset of ER luminal proteins and thus identified candidates for ER reflux cargos, not a definitive list (lines 279-281, 295-297 and 453).

3. Nucleoporin Relocalization (Figure 8). The observation that nucleoporins relocalize during ER stress is very interesting, but this section can be improved: It would be good to confirm by Western blot or fractionation that full-length nucleoporins are detected outside the nuclear envelope. Ubc6 is used as an ER marker in Figure 8B. Other markers for nuclear envelope or vesicles should be tested as well, and other nucleoporins should be analyzed. The authors should discuss if the puncta could be condensates, stress granules, or compartments for protein quality control, and not only products of NPC disassembly. Quantitative analysis of relocalization (e.g., how frequent it is among cells, or how fast it happens) would help. Since all conclusions are based on fluorescent fusions, Western blots should be used to confirm that the proteins remain full-length under stress and are not degraded or cleaved (see also below).

This comment contains several points, which we addressed as follows:

- 1) We analyzed Nsp1, Nup116 and Kap95, which we had imaged in Figure 8B and E, by Western blotting. We show that they remain intact during stress (new Figure S9B).
- 2) We had already before analyzed all 31 core nucleoporins by microscopy and summarized the results qualitatively in Table S6. We now quantified cytosolic clusters for six nucleoporin, three we had classified as puncta-forming, and three we had classified as non-puncta-forming. This confirmed the assessment of stress-induced puncta formation and also that the puncta-forming nucleoporins show some cytosolic clusters already at steady state (new Figure S9C).
- 3) We agree that a logical next step for characterizing the cytosolic nucleoporin puncta would now be a further analysis of their constituents, which could include testing other nuclear envelope proteins. However, we think that such an analysis would be outside the scope of this study. In our opinion, the nucleoporin experiments, without further extensions, convincingly demonstrate that dynamic organellar maps are capable of uncovering unanticipated protein localization changes, which was their purpose.
- 4) We explicitly mention in the discussion that the cytosolic nucleoporin puncta could be protein condensates. In addition, we cite two papers to direct the reader to the relevant literature on the possible roles of these structures in cellular stress responses and protein quality control (lines 505-506). We also explain that the puncta could result from NPC disassembly but also from NPC assembly defects (lines 508-515).

4. Fusion Protein Integrity. Please add a Western blot for selected tagged proteins (both control and stress). This would be useful to confirm that the fluorescence corresponds to full-length fusion proteins. This is especially important for membrane proteins that may be cleaved under stress.

We have added Western blots of Sil1-sfGFP-HDEL, Rtn1-mCherry, Prc1-sfGFP, Nsp1-mNeon, Nup116-mNeon and Kap95-mNeon under control and ER stress conditions. The full-length fusion

proteins are intact under all conditions except for Prc1-sfGFP. This luminal vacuole protein undergoes extensive cleavage at steady state so that free sfGFP is produced. This cleavage very likely reflects proteolytic processing in the vacuole. Cleavage is less extensive during ER stress, when a substantial portion of Prc1-sfGFP is retained in the ER and protected from vacuolar proteases. The case of Prc1-sfGFP nicely illustrates a key advantage of dynamic organellar mapping over microscopy of fluorescent fusion proteins, namely that it allows an analysis of native proteins. The data are shown in new Figures S6A, S6D, 6E and S9B and referred to in lines 291, 310-312, 361-367 and 432-433.

5. Membrane Contact Site Proteins. It would be interesting to look at known membrane contact site proteins, such as those described by Castro et al. (eLife, 2022). These proteins often link organelles and may not be well resolved in DOM PCA plots. Their behavior under ER stress could show changes in inter-organelle organization.

Exploring the potential remodeling of membrane contact sites during ER stress would certainly be interesting. We believe, however, that this deserves its own dedicated study and is outside the scope of the present paper. We here experimentally validated several expected and unexpected protein relocalization events to illustrate the power of Dynamic Organellar Maps. We summarized additional notable trends in the discussion, including a comment on contact site proteins (lines 517-530), to inspire future research.

6. Quantitative Microscopy Analysis. The microscopy images are helpful, but the analysis is mostly qualitative. Please add quantitative data — such as percentage of cells showing relocalization, or signal ratio between organelle and cytosol. This would help show the strength and consistency of the results.

We have now applied image analysis to quantify (1) cytosolic redistribution of Sil1 and Ero1 (also see point 2 above), (2) ER redistribution of eight vacuole, plasma membrane and Golgi proteins, and (3) cytosolic clustering of six nucleoporins (also see point 3 above). We have added these new data as Figures S6B, S7C, S7D, S8C, S8F and S9C. We had quantified the formation of cytosolic Rtn1 clusters previously (Papagiannidis et al, 2021).

Furthermore, we have added control experiments to test whether relocalization to the ER indeed reflected ER retention of newly synthesized protein molecules, as we had predicted. For this purpose, we placed Prc1-sfGFP and Gas3-sfGFP under the inducible GAL promoter and imaged the transport of newly synthesized protein without and during ER stress. Consistent with stress-induced ER retention, newly synthesized Prc1-sfGFP and Gas3-sfGFP were efficiently routed to the vacuole or the plasma membrane in unstressed cells but were retained to a substantial degree in the ER, likely because of misfolding. We have added these data as new Figures 6D, S7E and S8D and describe them in lines 358-361 and 383-384.

7. Abundance Changes in Relocalized Proteins. The analysis is mostly based on normalized localization profiles and does not include much about protein abundance. The authors say that most relocalized proteins do not change abundance, but actually 40% of them do in DTT, and 28% in tunicamycin. This is not a small fraction. It is important to demonstrate and discuss how abundance and localization changes are connected, and whether they might reflect coordinated responses.

Following the reviewer's suggestion, we have added a new Table S4D that combines the abundance and localization data for easy exploration. Among the proteins that change in both abundance and localization, there are groups of proteins that share certain properties but they do not appear to be particularly informative. For instance, 9/9 luminal ER proteins that redistribute to the cytosol also show increased abundance upon DTT treatment and are encoded by UPR target genes (according to Pincus et al, 2014). However, 7/9 luminal ER proteins that do not redistribute also show increased abundance. 12/17 vacuole proteins relocating upon DTT treatment also increase in abundance (70%) whereas only 26/51 of non-relocalizing vacuole proteins do (33%). However, the 12 upregulated and relocating proteins are not encoded by UPR target genes, so that it remains unclear how their upregulation may be coordinated. For now, we think it is best to provide the reader with the integrated view of the two levels of regulation in the new Table S4D but otherwise refrain from a detailed presentation of the data to not disrupt the flow of the paper.

8. Use of Tunicamycin. Although tunicamycin is included in the study, it is not discussed much. The authors should explain more clearly why it was used, and whether the localization changes it causes were confirmed or validated. It is unclear how tunicamycin-specific effects compare to DTT.

We thank the reviewer for prompting us to examine the impact of tunicamycin more thoroughly. Since tunicamycin and DTT may each have effects unrelated to ER stress, we followed the common practice in the field to use both drugs and focus on the cellular responses elicited by both drugs. We therefore did not investigate effects that clearly are specific for one drug, such as the tunicamycin-induced increase in overall in mitochondrial protein abundance. However, we have now tested experimentally whether certain protein localization changes are caused by both tunicamycin and DTT, as indicated by the organellar mapping data. Specifically, we have asked whether tunicamycin also triggers formation of Rtn1 puncta, redistribution of Prc1 and Gas1 to the ER, and clustering of Nsp1, Nup116 and Kap95 in the cytosol. In all cases, tunicamycin caused the same localization changes as DTT, although the effects were typically weaker. Thus, we believe that localization changes are generally shared by both stressors and we have added this information as new Figures S6E, S7B, S8B, S9A and S9E and describe it in lines 310, 357-358, 377-379 and 432-433.

9. Limitations of PCA. Many results are shown using PCA plots. PCA is good for visualization but may oversimplify complex data. The authors should discuss this limitation and explain if they validated the PCA results in another way.

We agree regarding the limitations of PCA plots and employed PCA plots primarily as a tool for data visualization. To make this clear, we have edited the first mention of PCA and now state that it represents "a simplifying visualization of multi-dimensional map data in only two dimensions" (lines 122-123). In some cases, we use protein shifts on PCA plots as an indicator that certain proteins undergo similar changes, for instance in Figure 5D (Rtn1, Yop1, Dpm1), 8A (nucleoporins) and S9D (Kap95, Nup116, Gle2). In these instances, we used microscopy for validation. We have edited the corresponding passages in the text to emphasize that the PCA plots only provided initial hints, which we then followed up by imaging experiments (lines 303, 411 and 430-431).

Minor points:

10. ER Membrane Remodeling and Terminology. Rtn1, Yop1, and Dpm1 relocalize into stress-induced puncta, and their sedimentation behavior changes. This suggests major remodeling of the ER membrane. The authors should make it clear that “membrane integrity” here means that the luminal content is retained, but does not exclude formation of subdomains or vesicles.

We explain in lines 312-317 that the structures containing Rtn1, Yop1 and Dpm1 could reflect the stress-induced formation of an ER subdomain or correspond to membrane vesicles. We have edited the text to make clear that such vesicles could form in intact cells or arise later through organelle fragmentation during cell lysis (lines 316-317). Regarding "membrane integrity", we unfortunately do not understand precisely where the reviewer wants us to modify the text because we do not use the term.

11. Redundancy in Figure 8A and 8C. Both panels show PCA plots of nucleoporins. It may be better to combine them, for example by using color to show which ones form puncta in Figure 8A.

It is true that Figure 8A and 8C could be combined. Nevertheless, we would prefer to keep them as they are because they make slightly different points (divergent behaviors among all nuclear envelope proteins versus divergent behaviors among nucleoporins). We feel that it will be easier for the readers to follow if these points are presented in two separate panels.

12. Mitochondrial Shifts in Figure 3A. Mitochondrial proteins show some shift, especially with tunicamycin, but this is not mentioned in the text. A short note on this would be helpful.

We have added a note concerning this shift, which indeed appears to be specific to tunicamycin (lines 205-208).

13. Interactive Tools Documentation. The interactive tools provided for data exploration are very nice and helpful. However, the documentation could be improved by including more detailed explanations of the different output types and what they represent. This would make the tools easier to understand and more accessible for users who are less familiar with DOM analysis.

We thank the reviewer for this comment, which prompted what we believe is a very helpful addition. We have expanded the documentation in Supplemental Databases 1 and 2 by including an extra tab with detailed ‘case studies’ of selected proteins. In this way, we aim to help the reader understand how the different output types can be interpreted. The examples in Supplemental Database 2 cover most of the proteins for which we also show microscopy images. We guide the reader to these examples in the results section (lines 244-246).